# Exploring the antimicrobial and antibiofilm potency of four essential oils against selected human pathogens using *in vitro* and *in silico* approaches

Kamal A. Qureshi[ID]1*, Adil Parvez2, Humaira Ismatullah3, Hanan Almahasheer[ID]4, Osamah Al Rugaie5*

1 Department of Pharmaceutics, College of Pharmacy, Qassim University, Buraydah, Saudi Arabia, 2 NextGen Life Sciences Pvt. Ltd., New Delhi, India, 3 School of Interdisciplinary Engineering & Sciences (SINES), National University of Sciences & Technology (NUST), Islamabad, Pakistan, 4 Department of Biology, College of Science, Imam Abdulrahman Bin Faisal University (IAU), Dammam, Saudi Arabia, 5 Department of Biology and Immunology, College of Medicine, Qassim University, Buraydah, Saudi Arabia

* ka.qurishe@qu.edu.sa (KAQ); o.alrugaie@qu.edu.sa (OAR)

## Abstract

Multi-drug-resistant (MDR) pathogens pose a significant global health challenge, underscoring the urgent need for novel antimicrobial agents with minimal toxicity to humans. This study investigated the *in vitro* and *in silico* antimicrobial and antibiofilm potentials of four essential oils (EOs): clove bud oil (CBO; *Syzygium aromaticum* L.), black seed oil (BSO; *Nigella sativa* L.), cinnamon bark oil (CNBO; *Cinnamomum zeylanicum*), and citronella oil (CTLO; *Cymbopogon nardus* L.), against 19 selected human pathogens, including MDR strains. Among the tested EOs, CBO, BSO, and CNBO exhibited the highest antibacterial activity against *Staphylococcus epidermidis*, with the mean zone of inhibition diameters (ZIDs) of 20.0 ± 0.2 mm, 46.0 ± 0.3 mm, and 32.0 ± 0.1 mm, respectively, at a concentration of 10 μL/disc, while CTLO displayed no antibacterial activity. CNBO demonstrated superior antifungal activity, with the mean ZIDs of 49.0 ± 0.3 mm and 36.0 ± 0.3 mm for *Candida albicans* and *Aspergillus niger*, respectively. Molecular docking analyses revealed robust interactions of key bioactive compounds—eugenol (EU) from CBO, thymoquinone (TQ) from BSO, cinnamaldehyde (CN) from CNBO, citronellal (CIT) and linalool (LIN) from CTLO—with microbial target proteins, substantiating their antimicrobial and antibiofilm potential. Notably, CTLO, despite limited *in vitro* activity, exhibited unique binding interactions *in silico*, suggesting potential niche applications. These findings underscore the translational potential of EOs as alternative antimicrobial therapies against MDR infections, particularly biofilm-associated infections, and highlight the need for further *in vivo* studies to validate their efficacy and safety.

## Introduction

MDR pathogens have emerged as a significant global threat, contributing to the high mortality rate worldwide [1]. Recent estimates suggest that antimicrobial resistance (AMR) accounts

**Data availability statement:** All relevant data are within the manuscript and its Supporting Information files.

**Funding:** The authors extend their appreciation to the Deputyship for Research & Innovation, Ministry of Education and, Saudi Arabia for funding this research work through the project number [QU-IF-1-4-1]. The authors also thank the technical support of Qassim University.

**Competing interests:** The authors have declared that no competing interests exist.

for nearly 1.27 million deaths annually, with projections indicating up to 10 million deaths per year by 2050 if no effective measures are implemented. Conventional antibiotics are increasingly failing to combat bacterial infections, creating a pressing global health burden, particularly in low- and middle-income countries with limited healthcare infrastructure. MDR pathogens, such as *Staphylococcus* spp., *Bacillus cereus* (*B. cereus*), pathogenic *Escherichia coli* (*E. coli*) strains, *Klebsiella* spp., *Pseudomonas* spp., *Salmonella* spp., *Shigella* spp., *Clostridium* spp., and *Aspergillus* spp., are among the most common contributors to the rise in fatality rates associated with drug-resistant infections [1–7].

The overuse and misuse of antibiotics have significantly accelerated the emergence of resistant bacterial strains, rendering many conventional treatments ineffective. Furthermore, synthetic antimicrobial compounds are often associated with adverse effects on human health [3,8]. Consequently, understanding the antibiotic resistance patterns of bacterial and fungal pathogens is critical to developing effective control strategies and selecting appropriate therapies for MDR infections.

Despite progress in antimicrobial research, there remains a substantial gap in identifying non-toxic, environmentally sustainable, and economically viable alternatives to antibiotics [9]. This gap highlights the need for exploring new antimicrobial solutions, particularly those derived from natural sources such as plants. Plants synthesize an array of secondary metabolites with potent antimicrobial properties, yet much of their potential remains untapped [3,9,10].

This study addresses this research gap by examining the antimicrobial and antibiofilm potential of four EOs—CBO, BSO, CNBO, and CTLO—against selected human pathogens, including MDR strains [10]. EOs are highly concentrated plant-derived products with documented antibacterial, antifungal, antiviral, antiparasitic, and antioxidant activities [3,8,10–28]. These natural products are purer and potentially more effective than traditional plant extracts [29].

## Rationale for EO Selection

The selection of these four EOs was guided by their traditional medicinal applications and reported antimicrobial properties:

- CBO: It is derived from the dried flower buds of *S. aromaticum* L. and has demonstrated antibacterial, antifungal, and antibiofilm activities, making it a potential candidate for treating bacterial infections resistant to conventional therapies [30].

- BSO: It is extracted from *N. sativa* L. seeds and has historically been used as a natural remedy in various cultures. Studies have highlighted its efficacy against bacterial and fungal pathogens [31].

- CNBO: It is sourced from the bark of *C. zeylanicum* L. Previous research has demonstrated that CNBO possesses significant antibacterial, antifungal, antiviral, and antibiofilm properties [32].

- CTLO: It is derived from the leaves of *Cymbopogon nardus* L., commonly known as citronella grass and is relatively understudied compared to other essential oils. While known for its calming properties in aromatherapy, emerging research suggests it possesses limited antimicrobial activity [33].

## Problem and study objective

The persistence of MDR pathogens and the inadequacy of existing treatments underscore the critical need for alternative therapies. This study aims to evaluate the antimicrobial and antibiofilm properties of four essential oils (EOs)—CBO, BSO, CNBO, and CTLO—using a

complementary approach that integrates *in vitro* and *in silico* methods. By coupling empirical evidence with molecular insights, this research provides a holistic understanding of these EOs' potential to combat MDR pathogens and offers a pathway for developing alternative therapeutic strategies.

## Materials and methods

### Test organisms

A total of 19 pathogenic organisms were used in this study, i.e., 17 bacteria: *Staphylococcus aureus* (*S. aureus*) ATCC 29213; *S. aureus*-CI (clinical isolate); *Methicillin-Resistant Staphylococcus aureus* (MRSA)-1; MRSA-2; *Staphylococcus saprophyticus* (*S. saprophyticus*) ATCC 43867; *Staphylococcus epidermidis* (*S. epidermidis*) ATCC 12228; *Streptococcus pyogenes* (*S. pyogenes*)-(A) ATCC 19615, *Streptococcus pneumoniae* (*S. pneumoniae*) ATCC 49619, *Enterococcus faecalis* (*E. faecalis*) ATCC 29212, *B. cereus* ATCC 10876, *E. coli* ATCC 25922, *Klebsiella pneumoniae* (*K. pneumoniae*) ATCC 27736, *Pseudomonas aeruginosa* (*P. aeruginosa*) ATCC 9027, *Salmonella typhimurium* (*S. typhimurium*) ATCC 13311, *Shigella flexneri* (*S. flexneri*) ATCC 12022, *Proteus vulgaris* (*P. vulgaris*) ATCC 6380, and *Proteus mirabilis* (*P. mirabilis*) ATCC 29906, and 02 fungi: *Candida albicans* (*C. albicans*) ATCC 10231, *Aspergillus niger* (*A. niger*) ATCC 6275. The bacterial strains were selected for their clinical relevance, especially as MDR pathogens associated with hospital-acquired infections. The fungal strains were chosen due to their prevalence in immunocompromised patients and their role in invasive fungal infections. All ATCC strains were sourced from Microbiologics® (St. Cloud, MN, USA), while the *S. aureus*-CI, MRSA-1, and MRSA-2 strains were obtained from King Saud Hospital (Unayzah, Saudi Arabia).

### Chemicals and reagents

Unless otherwise stated, all chemicals were purchased from Sigma-Aldrich (USA) or Oxoid Ltd. (UK). EOs were procured as follows:

- CBO: Loba Chemie Pvt. Ltd., Mumbai, India (CAS No: 8000-34-8, Eugenol content: ≥85% v/v).

- BSO: Al-Hussan Food Products Factory, Riyadh, Saudi Arabia (Purity: 100% v/v, Barcode No.: 6281102000134).

- CNBO and CTLO: Himani Herbal LLC, USA (Purity: 100% v/v, verified via vendor documentation and certificates of analysis).

To ensure reproducibility, EOs' purities were verified through vendor documentation and certificates of analysis provided by registered suppliers. EOs' dilutions were prepared in dimethyl sulfoxide (DMSO), which was included as a negative control in antimicrobial activity assays to confirm that it did not affect the stability or bioactivity of EOs throughout the study.

### In vitro antimicrobial profiles of EOs

**Preliminary antimicrobial activity.** The disc diffusion method [1,3,34,35] was used to evaluate the preliminary antimicrobial activities of the EOs. Modified Mueller-Hinton agar (MMHA) was used as a test medium [3]. Each test disc was prepared by dispensing 10 µL of pure EO. The inoculum turbidity of each organism was calibrated (optical density 0.08–0.12, equivalent to 0.5 McFarland standard) in sterile tryptic soy broth (TSB). On each

corresponding test medium plate, 100 μL of the respective calibrated inoculum was poured, and sterile swabs were used to spread the suspensions evenly. The prepared discs were placed on the inoculated plates. Then, bacterial and fungal plates were incubated at 35 ± 2°C for 24 and 48 h, respectively. Following incubation, the ZIDs were measured in millimeters (mm). Three separate sets of experiments were conducted. The results are presented as mean ± standard deviation (SD).

**Minimum Inhibitory Concentration (MIC) and Minimum Biocidal Concentration (MBC).** The MIC and MBC were determined using resazurin-based microbroth dilution and spot inoculation methods, respectively [3,35–37]. Stock solutions of EOs were prepared in DMSO at a concentration of 100 μL/mL, except for CTLO, which was prepared at 200 μL/mL. Subsequently, 200 μL of the stock solution was transferred to each well of column 1, whereas each well of columns 2-10 contained 100 μL of TSB only. In column 11, 200 μL of standardized inoculum suspension served as a negative control (NC), whereas in column 12, 200 μL of sterile TSB served as a sterility control (SC). A multichannel pipette was used to mix and transfer the EO from columns 1 to 10 to produce 100 μL/well of two-fold serial dilution.

Following Clinical and Laboratory Standards Institute (CLSI) standards, each test bacterium inoculum was made in TSB, and ODs were adjusted to 0.08-0.12 at $OD_{600}$ nm, resulting in $\sim 1 \times 10^8$ CFU/mL. The adjusted inocula were diluted in TSB by 1:100, resulting in approximately $1 \times 10^6$ CFU/mL. The inocula of the test fungi were prepared in potato dextrose broth (PDB) following CLSI guidelines; OD values were adjusted to 0.08-0.12 at $OD_{600}$, and the resulting stock suspensions were equal to $\sim 1 \times 10^6–5 \times 10^6$ CFU/mL for *C. albicans* and $4 \times 10^5–5 \times 10^6$ CFU/mL for *A. niger*. A working *C. albicans* suspension was prepared by diluting the stock suspension 1:100, followed by a 1:20 dilution in PDB, resulting in $\sim 5.0 \times 10^2$ to $2.5 \times 10^3$ cells/mL.

In contrast, a working *A. niger* suspension was prepared by diluting the stock suspension 1:50 in PDB, resulting in $\sim 0.8 \times 10^4$ to $1 \times 10^5$ spores/mL. 100 μL of adjusted microbial inocula were dispensed in all the wells of columns 1-10, resulting in $\sim 5 \times 10^5$ CFU/mL for bacteria, $\sim 2.5 \times 10^2$ to $1.25 \times 10^3$ CFU/mL for *C. albicans*, and $0.4 \times 10^4$ to $5 \times 10^4$ CFU/mL for *A. niger*. The preparation and dispensing of the OD-adjusted microbial inocula took 15 min. All the inoculated bacterial and fungal plates were incubated at 35 ± 2 °C for 24 and 48 h, respectively. Next, 30 μL of sterile resazurin (0.015%, w/v) solution was added to each well and re-incubated for 1-2 h. MIC refers to columns that remained blue during the incubation period. After a two-fold serial dilution of columns 1-10, CBO, BSO, and CNBO had 50-0.098 μL/mL EO concentrations, whereas CTLO had 100-0.195 μL/mL.

MBC was evaluated by inoculating the wells with concentrations from the MIC on sterile tryptic soy agar (TSA) and potato dextrose agar (PDA) plates for bacteria and fungi, respectively. MBCs had the lowest EO concentrations and did not generate isolated colonies of the test organisms on inoculated agar plates.

**Minimum Biofilm Inhibitory Concentration (MBIC) and Minimum Biofilm Eradication Concentration (MBEC).** MBIC is the lowest concentration of antimicrobial agent that prevents biofilm development in the tested organism. MBIC was performed on all microorganisms that were susceptible to EOs. The antibiofilm activity of EOs was assessed in a 96-well microtiter plate [3]. The inocula were prepared in TSB for bacteria and PDB for fungi at 0.5 MacFarland standard of (1–2 × 10^8 CFU/mL for bacteria, $1 \times 10^6 –5 \times 10^6$ CFU/mL for yeast, and $4 \times 10^5–5 \times 10^6$ CFU/mL for mold). A 100 μL aliquot of the optimized inoculum was placed in each test well of a 96-well plate. Next, 100 μL of EO solutions of varying concentrations were added to each test well. Therefore, the MIC, 2 × MIC, and 4 × MIC were selected as the final doses for MBIC testing. The wells designated "blank controls" (BC) were administered 200 μL sterile TSB/or PDB. The plates were incubated at 35 ± 2 °C for 24 h for bacteria and 48 h for fungi. After

incubation, the supernatant from each well was carefully removed by inverting the plates on a tissue paper bed. The plates were air-dried for 30 min, stained at room temperature for 30 min with 0.1% (w/v) crystal violet, and washed thrice with distilled water. Next, 200 μL of 95% ethanol was added to each test well to dissolve crystal violet. A microplate reader (xMarkTM Microplate Absorbance Spectrophotometer, Bio-Rad, USA) was used to measure absorbance at 650 nm. The MBICs had the lowest concentrations of EOs at which the absorbance was equal to or less than that of BC. All experiments were performed in triplicates. The mean of the three separate trials was calculated. Data are presented as μL/mL.

The minimum concentration of an antimicrobial agent required to eliminate the biofilm of the test organism is defined as the MBEC [3]. Each test well of a flat-bottom 96-well microtiter plate was inoculated with 200 μL of inoculum equivalent to 0.5, MacFarland standard ($1$–$2 \times 10^8$ CFU/mL for bacteria, $1 \times 10^6$–$5 \times 10^6$ CFU/mL for yeast, and $4 \times 10^5$–$5 \times 10^6$ CFU/mL for mold) of each test organism. For biofilm formation, the plates were incubated at $35 \pm 2$°C for 48 h for bacteria and 72 h for fungi. The contents of the test wells were removed after biofilm formation by inverting the plates over a tissue paper bed to remove non-adherent cells. Various concentrations of EOs, including the MIC, $2 \times$ MIC, and $4 \times$ MIC, were applied to different test wells (200 μL/well). The inoculated plates were re-incubated for 24 h at $35 \pm 2$°C. After incubation, the plates were inverted on a tissue bed to remove contents from the test wells. After 30 min of air drying, 200 μL of sterile TSB/PDB was added to each test well. Each test well received 30 μL 0.015% (w/v) resazurin dye. The plates were then re-incubated for another 1-2 h. MBEC values were obtained after reincubation by observing the color change from blue to pink. A column in which the color did not change (blue resazurin remained unchanged) was scored as the MBEC.

## Statistical analysis

The preliminary antimicrobial activities of the selected EOs were statistically analyzed using one-way analysis of variance (ANOVA) to determine the statistical differences in the mean antimicrobial values of the tested organisms. Post-hoc tests (Tukey's method) were performed to determine the significance of the interactions between the selected EOs for the tested organisms, where $p = 0.05$ was considered statistically significant. Statistical analyses were performed using SPSS version 26.0 (IBM, USA) [1,38].

## In silico molecular docking simulation

Five principal bioactive ingredients from these EOs were selected for molecular docking analysis based on a literature review (Table 1) [32,39–50].

**Target protein selection.** Twenty-three druggable target proteins, comprising bacterial and fungal proteins associated with biofilm formation, multi-drug resistance mechanisms, or virulence, were selected based on their clinical and therapeutic relevance. The protein structures were retrieved from the Protein Data Bank (PDB) (S3 Table) [55,56]. The selection criteria included their documented roles in antimicrobial resistance and virulence pathways and their prior investigation as therapeutic targets in both *in vitro* and *in vivo* studies. Proteins with well-characterized functions and those linked to pathways targeted by the bioactive compounds under investigation were prioritized.

To ensure the reliability of docking simulations, the 3D structures of the selected proteins were filtered based on crystallographic resolution, with only structures having a resolution of ≤ 2.0 Å included. Additionally, free R values were considered, and only structures with an R-free value not exceeding 0.05 (resolution/10) were included to ensure structural accuracy [57]. Of the 23 target proteins, 14 were bacterial, and 9 were fungal proteins.

**Table 1. Major bioactive constituents of EOs and their reported antimicrobial activities.**

| EOs | Major Constituents | Structures | Bioactivities | References |
|---|---|---|---|---|
| CBO | Eugenol | | • Antibacterial<br>• Antifungal | [39–42] |
| BSO | Thymoquinone | | • Antibacterial<br>• Antifungal | [43,44] |
| CNBO | Cinnamaldehyde | | • Antibacterial<br>• Antifungal | [32,45,46] |
| CTLO | Citronellal<br>Linalool | | • Antifungal | [51–54] |

The AlphaFold Protein Structure Database [58] was utilized for proteins with missing residues in their chains to supplement the 3D structures. Proteins with a high confidence score (pLDDT > 90) in AlphaFold were included, ensuring structural completeness and accuracy for docking studies.

**Docking protocol validation.** A thorough validation process was conducted to ensure the reliability and accuracy of the docking protocol. Re-docking experiments were performed by docking the co-crystallized ligands of the selected target proteins back into their binding sites. The root-mean-square deviation (RMSD) between the predicted and crystallographic poses was calculated, with RMSD values ≤ 2.0 Å indicative of successful validation.

Additionally, benchmark ligands with well-documented binding affinities were included in the study to validate the docking results' consistency with experimental data. This step ensured that the docking protocol was robust and produced results aligned with established experimental evidence, reinforcing the reliability of the docking simulations used in this study.

**Docking simulations.** Molecular docking simulations of the selected ligands were performed using the Molecular Operating Environment (MOE) V2022.02 software tool [59–64]. First, the protein structures of the targets were prepared by removing co-crystallized ligands and water molecules. The 3D structures were corrected for missing chain residues by adding hydrogen atoms and assigning protonation states by generating conformers. Protonation was performed by adding 80% solvation at pH 7 and 300 K (default parameters). Subsequently, energy minimization was performed by computing the forcefield Amber10:EHT in the MOE [65–67]. The active site of the target proteins was defined by the residues containing the co-crystallized ligand or was located using the site finder tool in the MOE. The largest cavity was selected as the binding site based on a higher Propensity for Ligand Binding (PLB) score.

The chemical structures of the ligands were retrieved from the PubChem database [68,69], and their respective PubChem identification numbers are 3314 for EU, 10281 for TQ, 637511 for CN, 7794 for CIT, and 6549 for LIN [3,38]. The 3D structures of the ligands were built through the Builder application by placing the canonical SMILES of the ligands, followed

by energy minimization using the MMFF94X force field (with parameters 0.0001 gradient, 10, and 12 cut-off values), and partial charges were applied [70] in MOE. Furthermore, the induced fit docking protocol [71] was applied using the default placement method (Triangle Matcher) and a scoring function (London dG). Molecular docking simulations were performed with 30 poses generated per ligand, followed by refinement with a refinement number of 5. Based on the lowest binding energy value, the best pose for each ligand was selected for detailed protein-ligand interaction analysis.

# Results

## Preliminary antimicrobial activity

Preliminary antimicrobial activity showed that CBO, BSO, CNBO, and CTLO had substantial antimicrobial potential against the tested pathogens, except for CTLO, which only had antifungal activity at 10 μL oil/disc. DMSO, the negative control (N), exhibited no antimicrobial activity against any tested microorganisms at 10 μL/disc; hence, diluting EOs for MIC, MBC, MBIC, and MBEC is acceptable.

The antibacterial activity revealed that CBO, BSO, and CNBO had mean ZIDs in ranges of 9.0-20.0 mm, 17.0-46.0 mm, and 13.0-32.0 mm, respectively, against the tested bacteria. Similarly, the mean ZIDs for the tested fungi were in ranges of 23.0-32.0 mm, 10.0-11.0 mm, 36.0-49.0 mm, and 8.0-13.0 mm for CBO, BSO, CNBO, and CTLO, respectively. Furthermore, the results showed that *S. epidermidis* ATCC 12228 was the most susceptible test bacterium for CBO, BSO, and CNBO, with mean ZIDs of 20.0 ± 0.2 mm, 46.0 ± 0.3 mm, and 32.0 ± 0.1 mm, respectively. Furthermore, results showed that *P. aeruginosa* ATCC 9027 is the least susceptible test bacterium for CBO and CNBO, with mean ZIDs of 9.0 ± 0.2 mm and 13.0 ± 0.3 mm, respectively. In contrast, BSO did not affect Gram-negative bacteria at a concentration of 10 μL oil/disc (Figs 1–2).

The antifungal activity results showed that CNBO exhibited the most potent antifungal activity against the tested fungal strains, with mean ZIDs of 49.0 ± 0.3 mm and 36.0 ± 0.3 mm for *C. albicans* ATCC 10231 and *A. niger* ATCC 6275, respectively. CBO and BSO demonstrated moderate antifungal activity, with mean ZIDs of 32.0 ± 0.3 mm and 11.0 ± 0.3 mm against *A. niger* ATCC 6275, and 23.0 ± 0.3 mm and 10.0 ± 0.3 mm against *C. albicans* ATCC 10231, respectively. CTLO exhibited weaker antifungal activity, with mean ZIDs of 13.0 ± 0.3 mm against *C. albicans* ATCC 10231 and 8.0 ± 0.2 mm against *A. niger* ATCC 6275 (Figs 1–2).

The selective antifungal activity of CTLO may be attributed to its major components, CIT and LIN, which are known to disrupt fungal cell membranes by altering membrane fluidity or targeting fungal-specific sterols like ergosterol. However, these components might lack efficacy against bacterial membranes, which possess structural differences such as peptidoglycan and lipopolysaccharides that require different disruption mechanisms. In contrast, CNBO's superior antifungal activity can be linked to its high CN content, a phenolic compound with strong antimicrobial properties that target both bacterial and fungal membranes. These findings suggest that the chemical composition of EOs plays a crucial role in determining their spectrum of antimicrobial activity. Thus, based on the preliminary antimicrobial activity results, we conclude that CNBO is the most potent, CBO and BSO are moderate, and CTLO is the least potent antimicrobial essential oil.

## MIC, MBC, MBIC, and MBEC

The MIC values for CBO, BSO, and CNBO against the tested bacteria ranged from 1.56 to 25.0 μL/mL, 0.78 to 3.125 μL/mL, and 0.78 to 6.25 μL/mL, respectively. The corresponding MBC values were between 3.125 and 50.0 μL/mL, 1.56 and 6.25 μL/mL, and 1.56 and 12.50

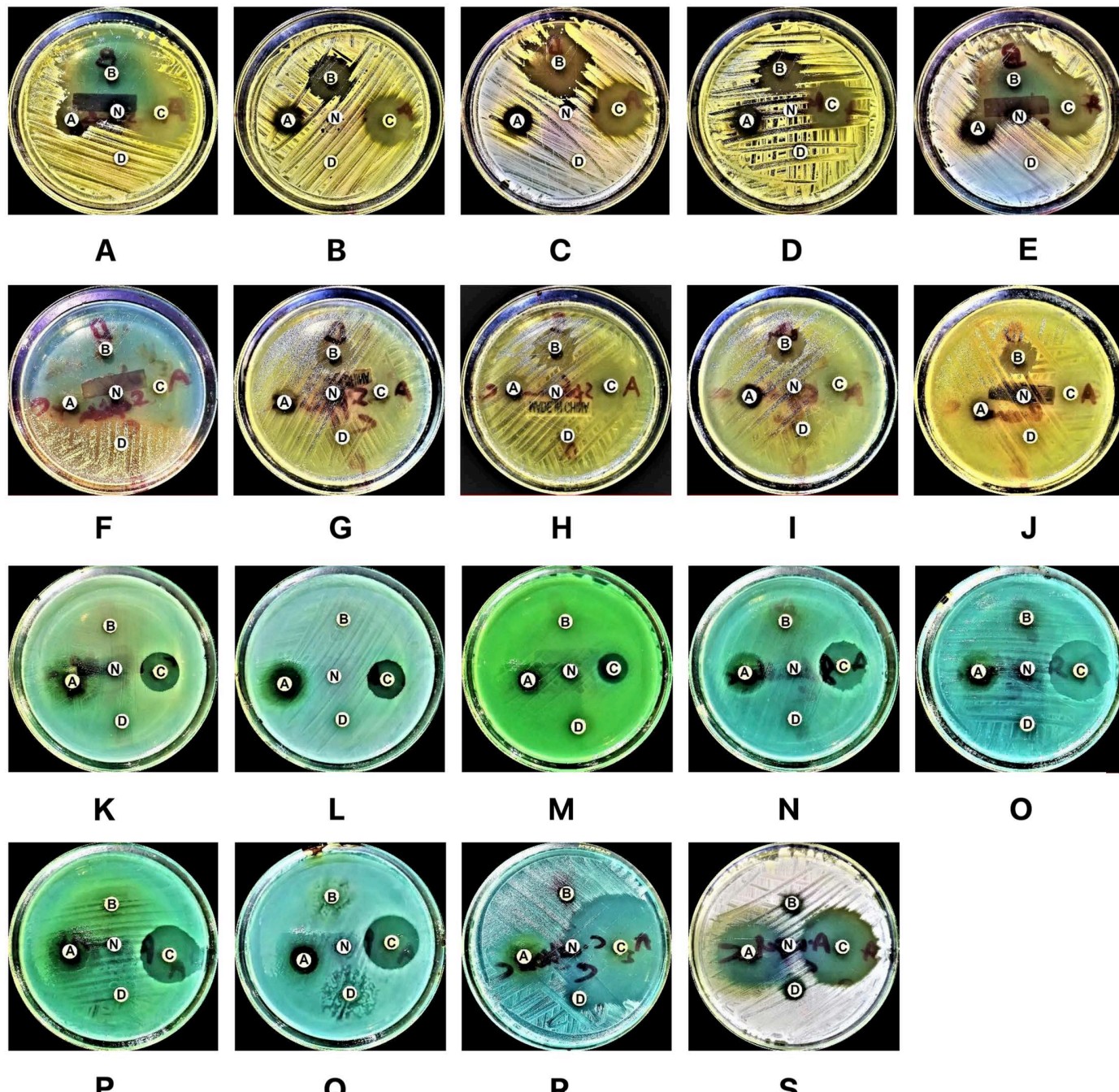

**Fig 1. (A–S): Preliminary antimicrobial activity of EOs against tested microorganisms.** This figure illustrates the antimicrobial activity of EOs, where disc A represents CBO, disc B represents BSO, disc C represents CNBO, disc D represents CTLO, and disc N represents the negative control (DMSO). The corresponding figures depict the following microorganisms: **A** = *S. aureus* ATCC 29213, **B** = *S. aureus*-CI, **C** = MRSA-1, **D** = MRSA-2, **E** = *S. saprophyticus* ATCC 43867, **F** = *S. epidermidis* ATCC 12228, **G** = *S. pyogenes*-(A) ATCC 19615, **H** = *S. pneumoniae* ATCC 49619, I = *E. faecalis* ATCC 29212, **J** = *B. cereus* ATCC 10876, **K** = *E. coli* ATCC 25922, **L** = *K. pneumoniae* ATCC 27736, **M** = *P. aeruginosa* ATCC 9027, **N** = *S. typhimurium* ATCC 13311, **O** = *S. flexneri* ATCC 12022, **P** = *P. vulgaris* ATCC 6380, **Q** = *P. mirabilis* ATCC 29906, **R** = *C. albicans* ATCC 10231, and **S** = *A. niger* ATCC 6275. Antimicrobial activity is demonstrated by clear zones of inhibition surrounding each disc.

**Preliminary Antimicrobial Activity of Essential Oils**

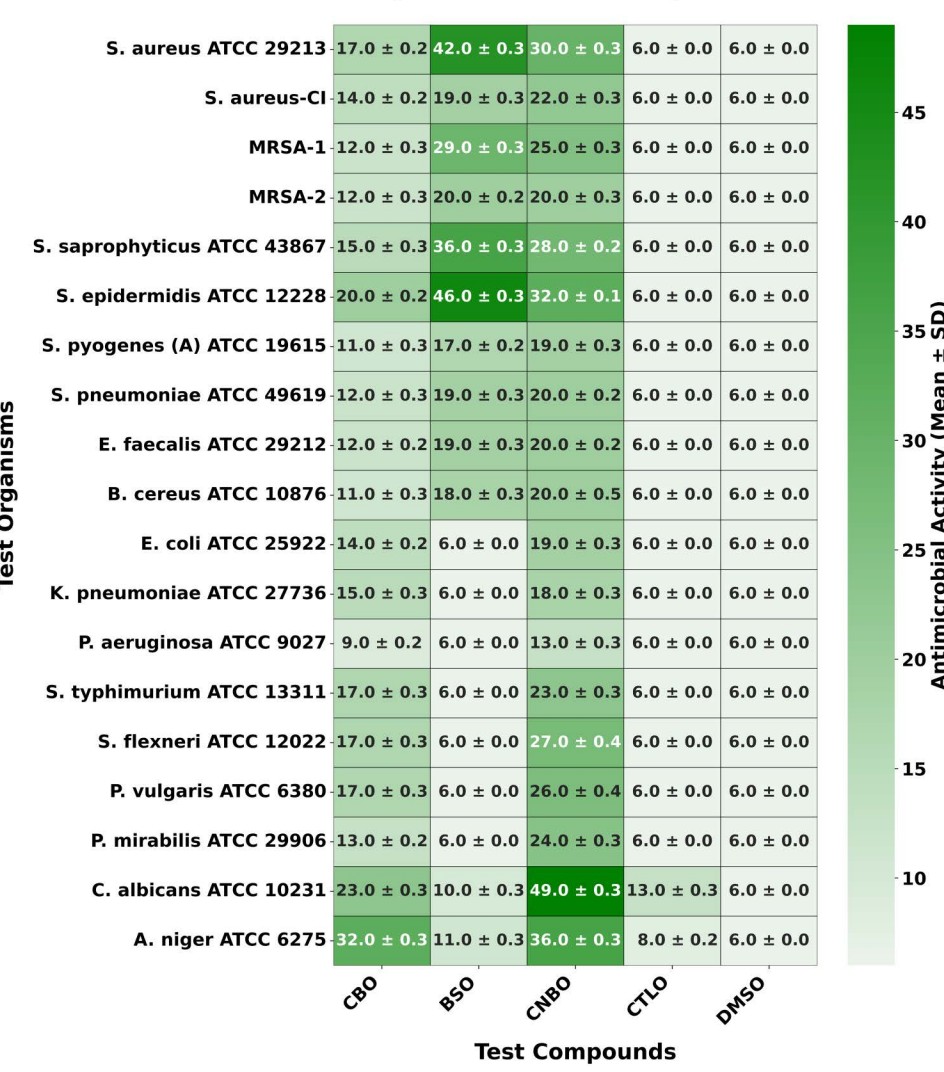

**Fig 2. Heatmap illustrating the preliminary antimicrobial activity of EOs and DMSO.** This figure presents a heatmap showcasing the preliminary antimicrobial activity of EOs and the negative control, DMSO. A value of 6.0 ± 0.0 indicates no zone of inhibition and, therefore, no antimicrobial activity.

µL/mL. These findings are consistent with theoretical expectations that EOs with potent antimicrobial compounds, such as EU in CBO, TQ in BSO, and CN in CNBO, would exhibit significant inhibitory and bactericidal activities.

For the fungal strain *C. albicans* ATCC 10231, the MIC values of CBO, BSO, CNBO, and CTLO were found to be 0.195 µL/mL, 1.56 µL/mL, 0.195 µL/mL, and 1.56 µL/mL, respectively, with corresponding MBC values of 0.39 µL/mL, 3.125 µL/mL, 0.39 µL/mL, and 3.125 µL/mL. Similarly, for *A. niger* ATCC 6275, the MIC values for CBO, BSO, CNBO, and CTLO were 0.195 µL/mL, 1.56 µL/mL, 0.195 µL/mL, and 0.78 µL/mL, respectively, while the MBC values were 0.39 µL/mL, 3.125 µL/mL, 0.39 µL/mL, and 1.56 µL/mL, respectively (Figs 3–6). These results align with the theoretical framework, suggesting that oils rich in antifungal components, such as EU in CBO, TQ in BSO, CN in CNBO, and CIT in CTLO, are effective at low concentrations against fungal pathogens.

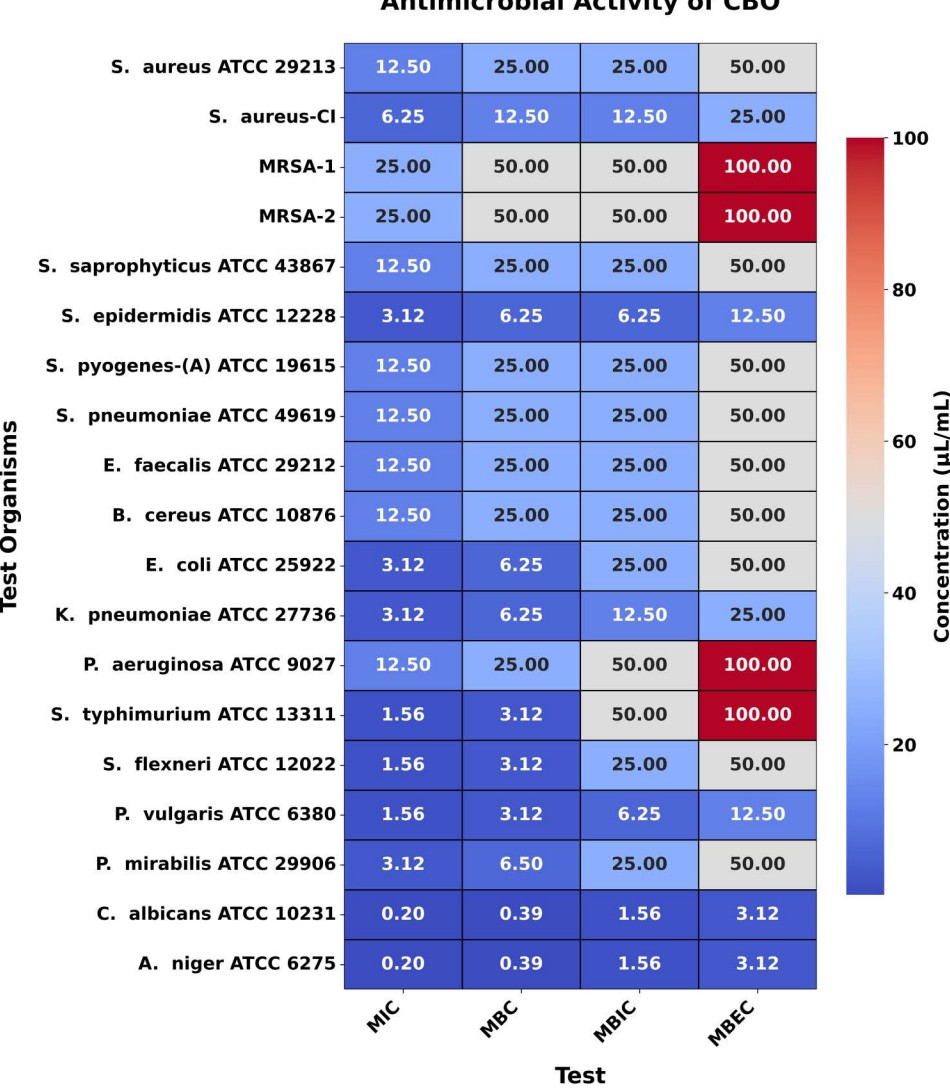

**Fig 3. Heatmap illustrating the antimicrobial activity of CBO.** This figure presents a heatmap of the antimicrobial activity of CBO against test organisms, displaying MIC, MBC, MBIC, and MBEC values in µL/mL. The color gradient reflects the concentration, with darker shades indicating higher values while lower values represent greater potency.

Moreover, the results of MBIC and MBEC demonstrate that these EOs effectively prevent biofilm formation and eradicate established biofilms. This is particularly relevant in clinical settings, where biofilms contribute to the persistence and resistance of infections. The MBIC and MBEC values reinforce the understanding that biofilms require higher concentrations of antimicrobial agents, and the oils tested in this study effectively address this challenge.

## Statistical analysis

One-way ANOVA analyses revealed that there is a statistically significant difference ($p < 0.05$) in the mean antimicrobial values between the groups of tested microbial strains for CBO, BSO, CNBO, and CTLO, i.e., CBO; $F_{(18, 38)} = 1042.982$, $p = 0.000$, BSO; $F_{(18, 38)} = 8652.648$, $p = 0.000$, CNBO; $F_{(18, 38)} = 1796.748$, $p = 0.000$, CTLO; $F_{(18, 38)} = 1028.889$, $p$

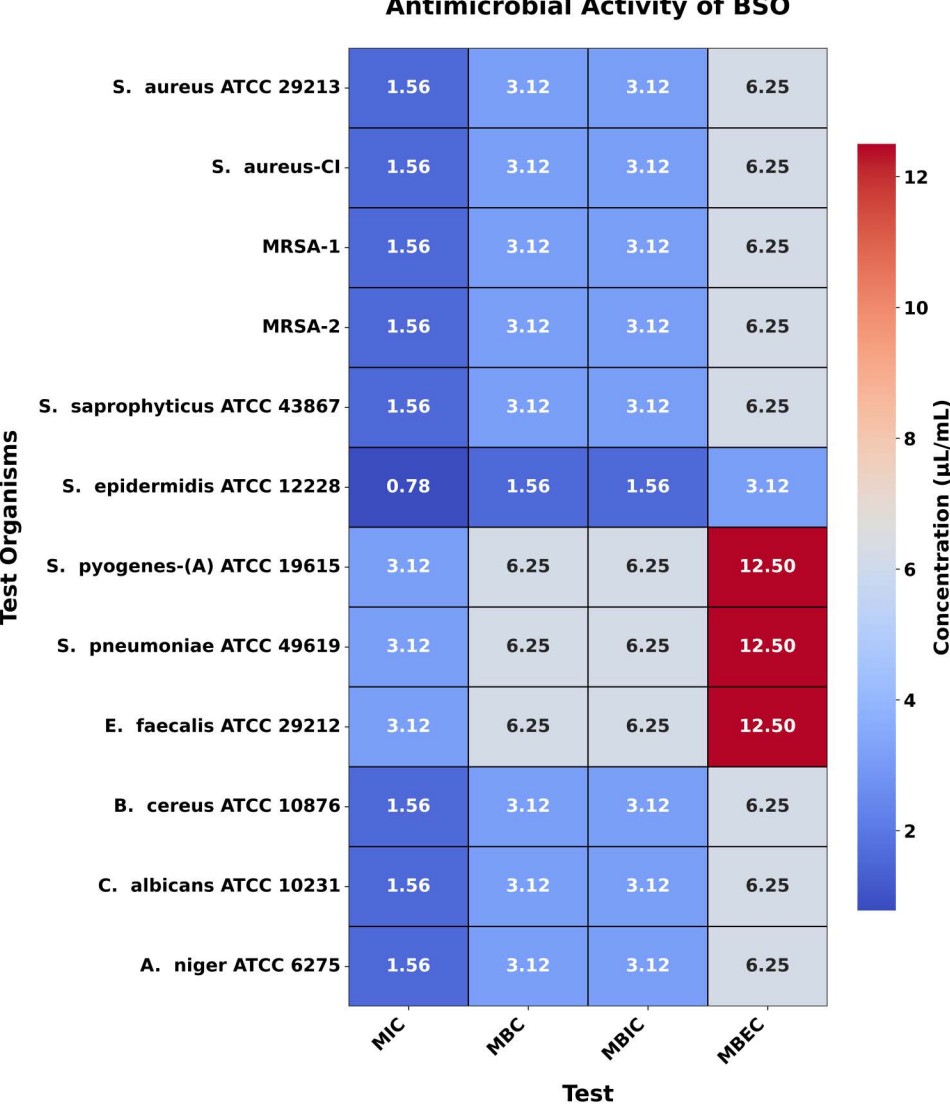

**Fig 4. Heatmap illustrating the antimicrobial activity of BSO.** This figure presents a heatmap of the antimicrobial activity of BSO against test organisms, displaying MIC, MBC, MBIC, and MBEC values in μL/mL. The color gradient reflects the concentration, with darker shades indicating higher values while lower values represent greater potency.

= 0.000 (S2 Table). Post-hoc analyses revealed that there is a statistically significant difference ($p < 0.05$) in the mean antimicrobial values between the tested compounds, except for BSO and CNBO for MRSA-2 and BSO and CTLO for the Gram-negative bacteria, i.e., BSO and CNBO for MRSA-2; $p = 1.000$ and BSO and CTLO for all the Gram-negative bacteria; $p = 1.000$ (S3 Table).

## Molecular docking simulation

Molecular docking analysis focused on the interactions between the five major bioactive compounds from the selected EOs and 23 target proteins associated with the selected pathogens. Across all key target proteins, the bioactive compounds demonstrated significant overlap with critical residues, particularly those involved in catalytic or binding mechanisms, as observed in

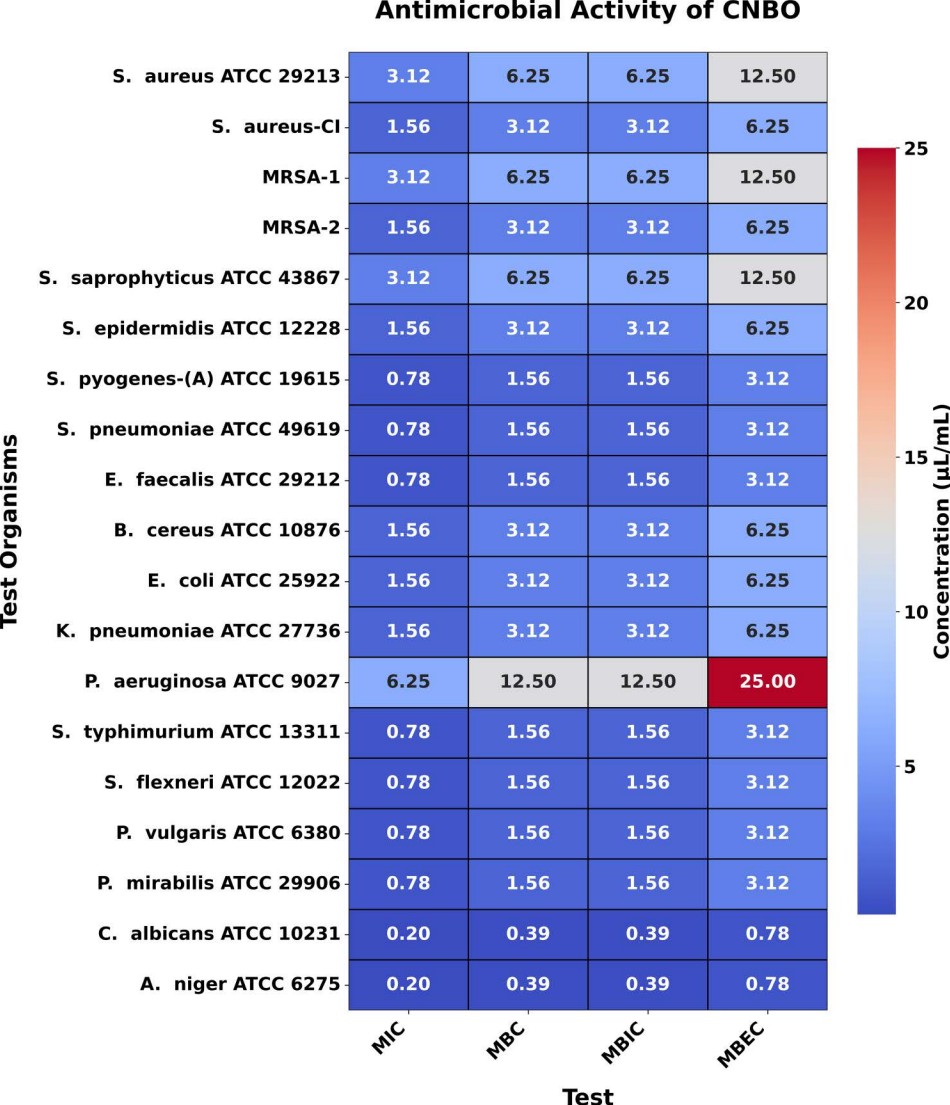

**Fig 5. Heatmap illustrating the antimicrobial activity of CNBO.** This figure presents a heatmap of the antimicrobial activity of CNBO against test organisms, displaying MIC, MBC, MBIC, and MBEC values in µL/mL. The color gradient reflects the concentration, with darker shades indicating higher values while lower values represent greater potency.

the co-crystal or standard ligand-docked structures. This overlap underscores the accuracy of our docking results and reinforces the importance of these interactions in maintaining protein function. The binding energies and specific interactions of each selected bioactive compound with their respective target proteins are shown in Figs 7 and Table 2. Below is the detailed breakdown of molecular docking results:

a) **EU**

- **5MM8:** EU exhibited a robust binding profile with a docking score of -10.48 kcal/mol, facilitated by key hydrogen bonds and extensive non-covalent interactions. These interactions significantly contribute to the stability and specificity of the ligand-protein complex.

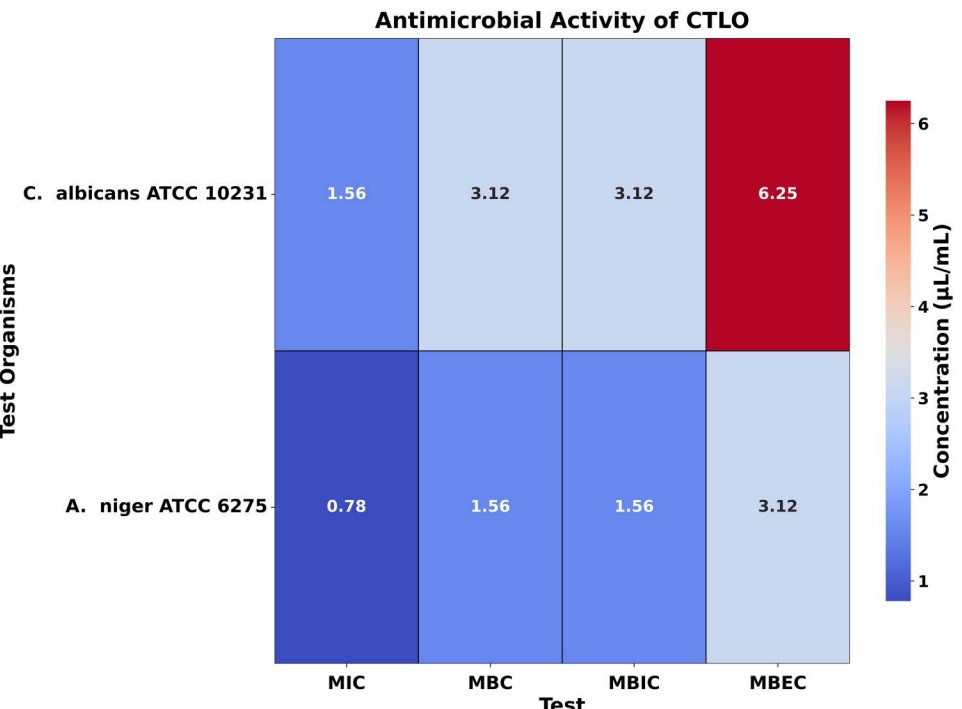

**Fig 6. Heatmap illustrating the antimicrobial activity of CTLO.** This figure presents a heatmap of the antimicrobial activity of CTLO against test organisms, displaying MIC, MBC, MBIC, and MBEC values in µL/mL. The color gradient reflects the concentration, with darker shades indicating higher values while lower values represent greater potency.

Notably, an unfavorable interaction with Ser132 highlights a potential opportunity for ligand optimization to enhance binding efficiency. Furthermore, the hydrophobic pocket, reinforced by alkyl and π-alkyl interactions, provided additional stabilization to the ligand within the active site (Figs 7-8A, 9, and Table 2).

- **5UIV:** EU demonstrated strong interaction with a docking score of -10.66 kcal/mol, driven by critical hydrogen bonds, π-cation/anion interactions, and extensive van der Waals forces. Electrostatic and hydrophobic forces (via π-cation and π-alkyl interactions) were pivotal in maintaining binding stability. This interaction profile suggests that EU effectively binds to the 5UIV protein, highlighting these residues as critical for potential inhibitor design or the optimization of EU derivatives (Figs 7-8B, 10, and Table 2).

b) **TQ**

- **5MM8:** TQ exhibited stable and specific binding with a docking score of -9.32 kcal/mol, facilitated by key hydrogen bonds with Glu151 and Asn154, as well as significant alkyl and π-alkyl interactions with hydrophobic residues such as Pro120, Val4, and Phe149. The presence of van der Waals interactions with numerous residues further enhanced the stability of the ligand-protein complex, making this interaction robust. These diverse interaction types highlight a promising binding affinity, suggesting TQ's potential as an effective inhibitor or modulator of the target protein's function (Figs 7-8C, 11, and Table 2).

- **5JPF:** TQ showed a strong interaction profile with a docking score of -9.09 kcal/mol, stabilized by key hydrogen bonds with residues such as Asn289, Arg386, Tyr437, and Arg261.

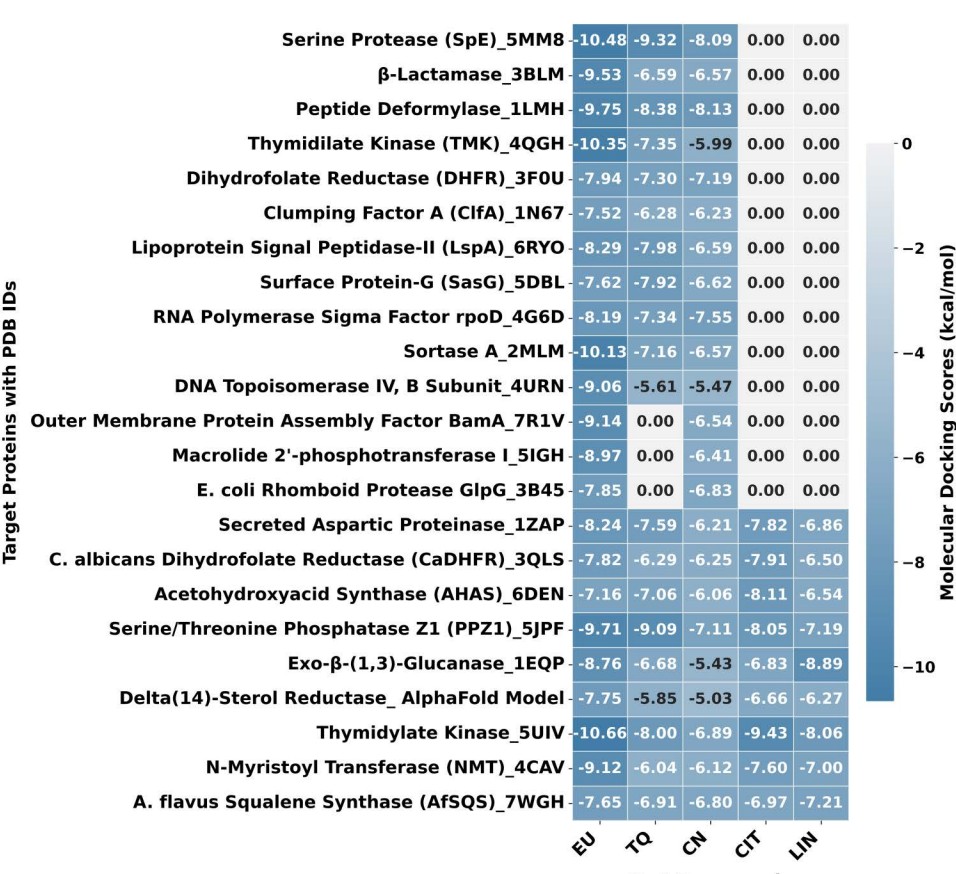

**Fig 7. Heatmap illustrating the molecular docking scores of active compounds from EOs.** This figure shows a heatmap of molecular docking scores (kcal/mol) for active compounds from EOs against target proteins. Lower scores indicate stronger binding. Target proteins and compounds are listed on the axes, and the color gradient represents the range of docking scores.

The addition of π-cation and π-π interactions further contributed to the electrostatic and hydrophobic stability of the complex. The involvement of multiple van der Waals interactions ensured the proper orientation of the ligand within the binding pocket. This combination of interactions suggests that TQ binds effectively to the 5JPF protein, making it a strong candidate for further exploration in drug design targeting this protein (Figs 7-8D, 12, and Table 2).

c) **CN**

- **1LMH:** CN formed a stable complex with the 1LMH protein, achieving a docking score of -8.13 kcal/mol. Key covalent bonds with Gln153 and Asp157 and non-covalent π-π T-shaped and π-alkyl interactions with residues such as Arg124 and His125 mediated this binding. Additionally, van der Waals interactions contributed to the stability and robustness of the ligand-protein complex. These findings suggest CN has a strong affinity for 1LMH, making it a promising candidate for further drug design exploration and biological activity studies (Figs 7-8E, 13, and Table 2).

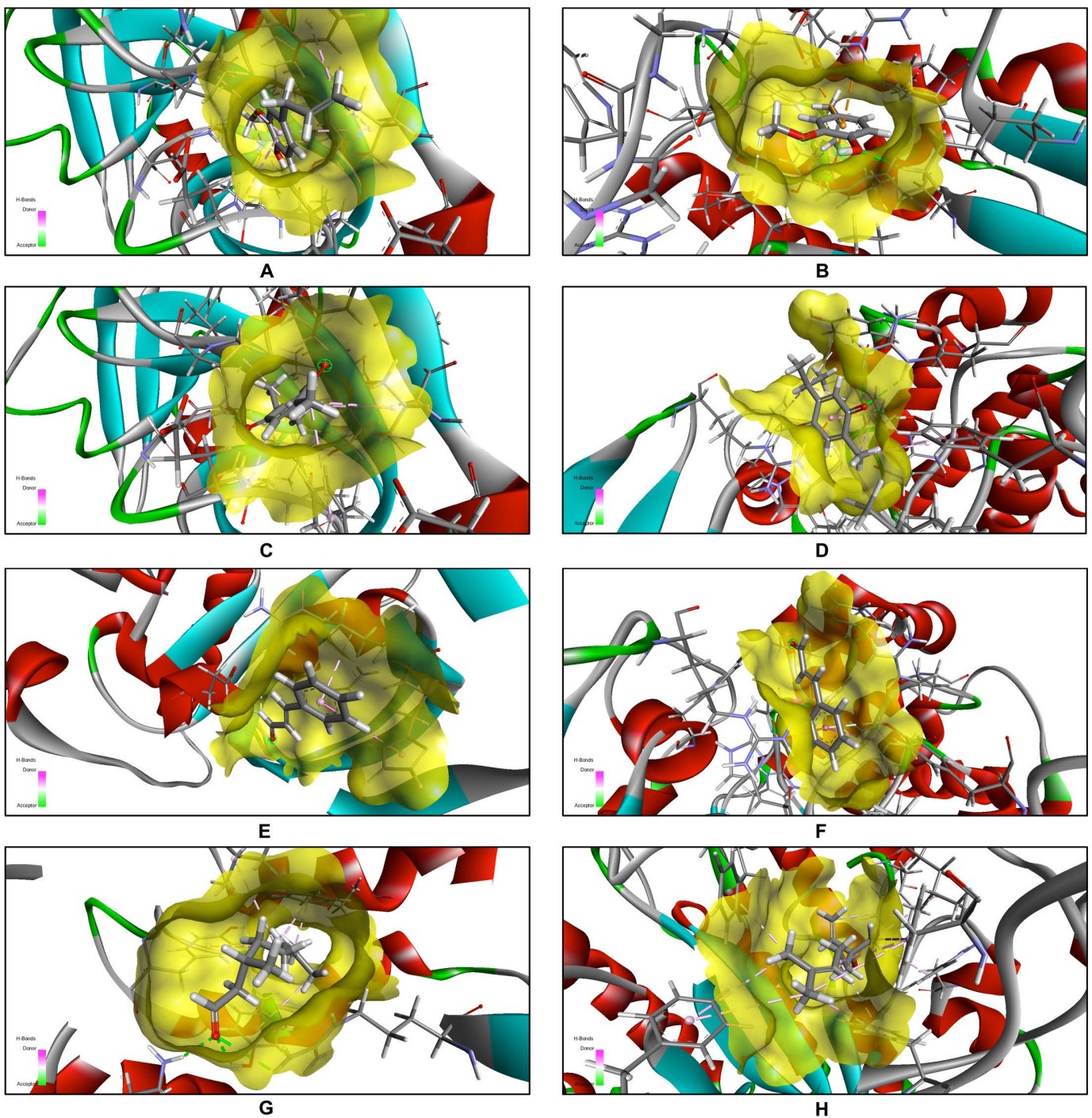

**Fig 8. (A–H): 3D ligand-protein interactions in active-site pockets.** These figures (**A–H**) depict the 3D ligand-protein interactions within the active-site pockets of target proteins. The corresponding ligand-protein complexes are as follows: **Fig A** = 5MM8-EU, **Fig B** = 5UIV-EU, **Fig C** = 5MM8-TQ, **Fig D** = 5JPF-TQ, **Fig E** = 1LMH-CN, **Fig F** = 5JPF-CN, **Fig G** = 5UIV-CIT, and **Fig H** = 1EQP-LIN. All 3D and 2D docked images were generated and visualized using Biovia Discovery Studio Visualizer 2021 (https://discover.3ds.com).

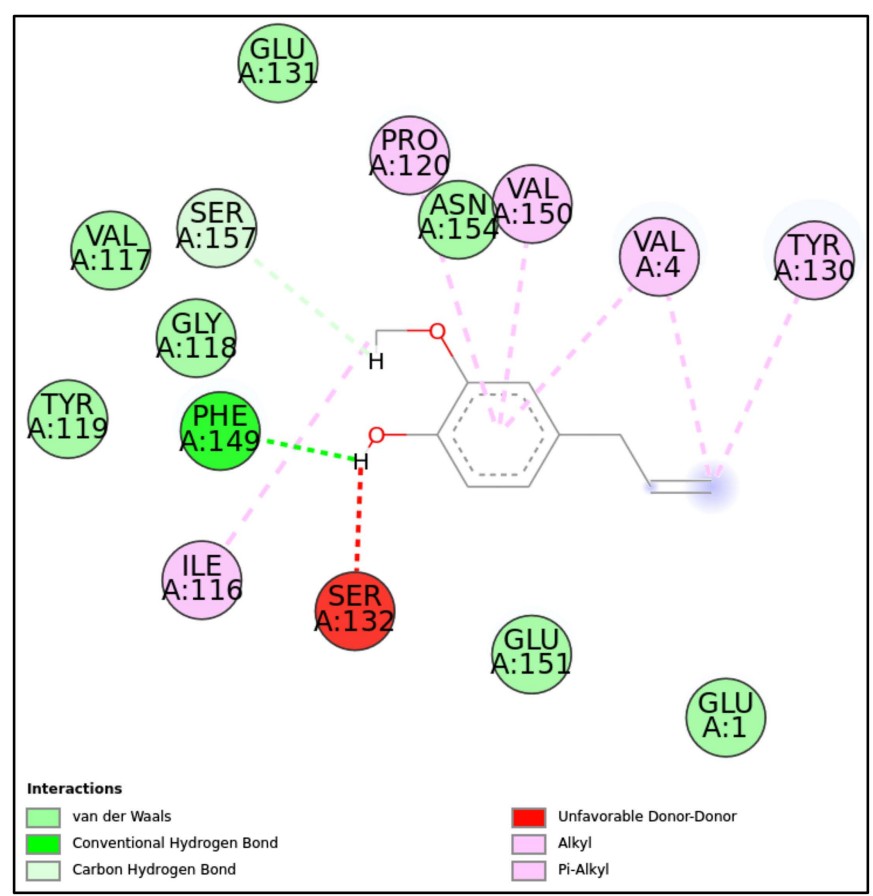

**Fig 9. 2D molecular interactions between EU and 5MM8 (PDB).** 2D interactions between EU and 5MM8, highlighting hydrogen bonds, van der Waals forces, alkyl, and π-alkyl interactions.

- **5JPF:** The binding of CN to the 5JPF protein was primarily driven by π-cation interactions (with Arg386) and π-π T-shaped interactions (with His231), with a docking score of -7.11 kcal/mol. Notably, a covalent interaction with Tyr437 suggests a strong and potentially irreversible binding event. Extensive van der Waals interactions provided additional stability and ensured proper ligand orientation within the binding pocket. The absence of conventional hydrogen bonds indicates that non-covalent interactions are pivotal in maintaining the ligand's binding. This interaction profile highlights CN as a stable and potentially strong binder to 5JPF, mainly due to its covalent interaction with Tyr437 (Figs 7-8E, 14, and Table 2).

d) **CIT**

- **5UIV:** CIT formed a stable complex with the 5UIV protein, achieving a docking score of -9.43 kcal/mol. This binding was mediated by critical hydrogen bonds with Tyr100, Gln167, and Arg92, which are essential for ligand specificity. Additionally, the combination of hydrophobic alkyl and π-alkyl interactions with residues such as Phe67, Tyr161, and Leu51 contributed significantly to the ligand's stability within the binding pocket. Van der Waals interactions further enhanced the orientation and positioning of the ligand. The overall interaction profile highlights CIT's strong binding affinity to the 5UIV protein, making it a

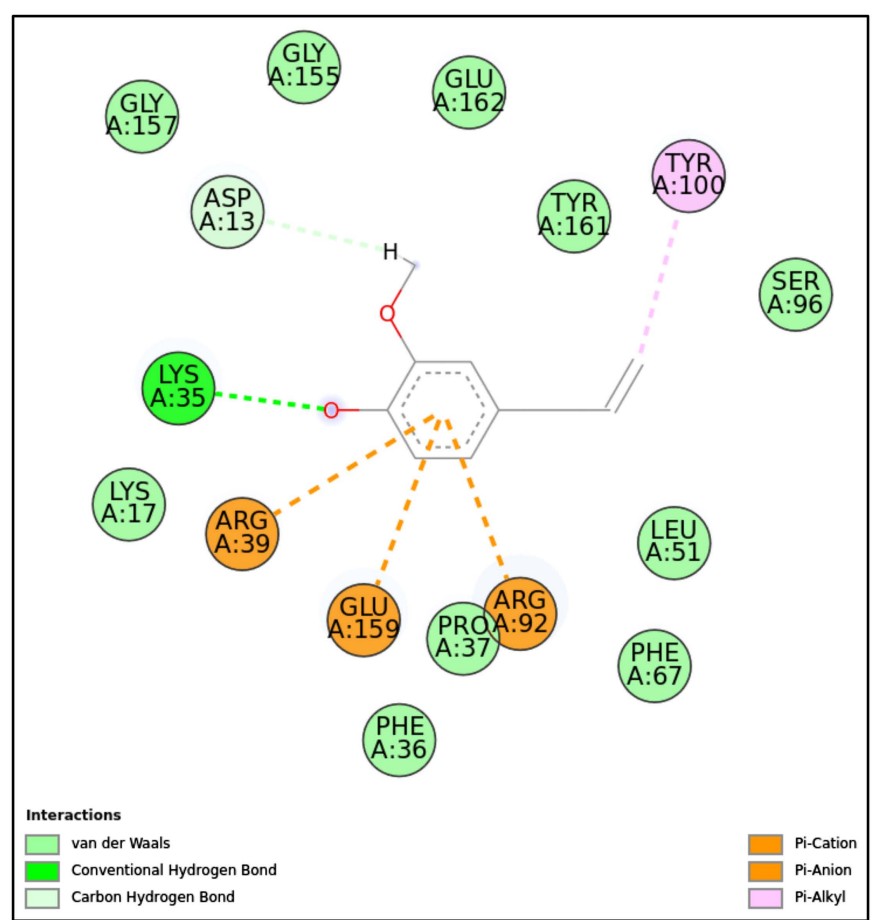

**Fig 10.** 2D **molecular interactions between EU and 5UIV (PDB).** 2D molecular interactions between EU and 5UIV, showing hydrogen bonds, van der Waals forces, π-cation, π-anion, and π-alkyl **interactions.**

promising candidate for further exploration in drug design and studies targeting this protein system (Figs 7-8F, 15, and Table 2).

e) **LIN**

- **1EQP:** LIN interacted with the 1EQP protein with a docking score of -8.89 kcal/mol, primarily mediated through π-alkyl and van der Waals interactions. The absence of conventional hydrogen bonds and covalent interactions suggests that LIN's binding relies heavily on hydrophobic forces, particularly involving aromatic residues such as Phe144, Trp373, and Tyr29. Van der Waals interactions further contributed to the stability and proper positioning of the ligand within the binding pocket. This interaction profile indicates a moderate binding affinity driven by hydrophobic and weak non-covalent forces, making LIN a potential candidate for further studies targeting this protein system (Figs 7-8F, 16, and Table 2).

The 3D docking interactions (Fig 8) and 2D molecular interactions (Figs 9–16) visually demonstrate the bioactive compounds' contribution to their antimicrobial properties by targeting essential enzymes in pathogens, thereby inhibiting their growth and survival.

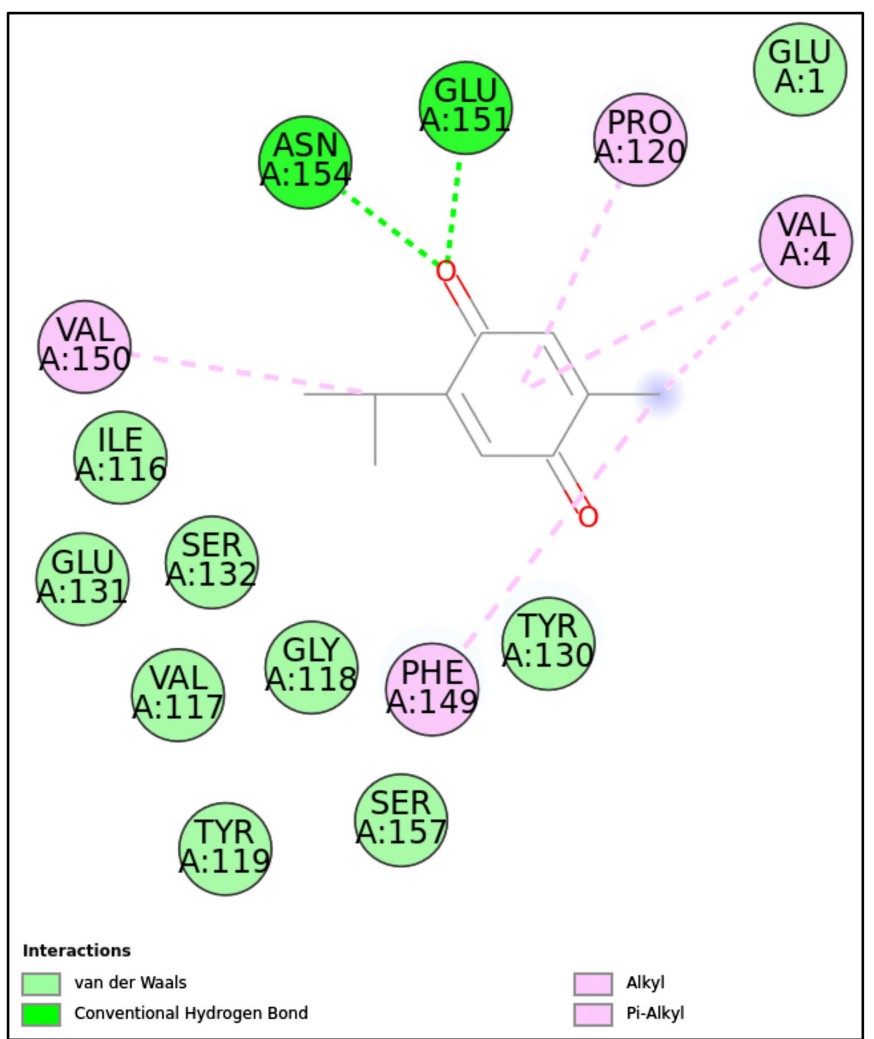

**Fig 11.** 2D **molecular interactions between TQ and 5MM8 (PDB).** 2D molecular interactions between TQ and 5MM8, showing van der Waals forces, conventional hydrogen bonds, alkyl, and π-alkyl interactions.

We validated our computational findings by comparing the binding interactions of the selected bioactive compounds with standard or co-crystallized ligands in their respective protein targets. For the 5MM8 target protein, benzamidine (BMD); for the 5UIV target protein, thymidine-5'-phosphate (TMP); for the 5JPF target protein, microcystin-LR (MLR); for the 1LMH target protein, actinonin (ACT); and for the 1EQP target protein, laminaran (LAM) were used as standard ligands (S4 Table). The selection criteria for these ligands were based on either co-crystallization data or literature references. The results of this analysis are presented below [72–76]:

a) **BMD with 5MM8:** BMD demonstrates robust binding with the 5MM8 protein through conventional hydrogen bonds involving Ser157 and Tyr130, significantly contributing to stability. Hydrophobic interactions, including π-alkyl interactions with Pro120, Val4, and Val150, further enhance the ligand's fit within the active site. Additionally, van der Waals interactions with Glu151, Ile116, Gly118, Val117, Tyr119, Glu131, Ser132, Lys5, and Phe149 stabilize the ligand within the binding pocket. Although an unfavorable donor-donor interaction

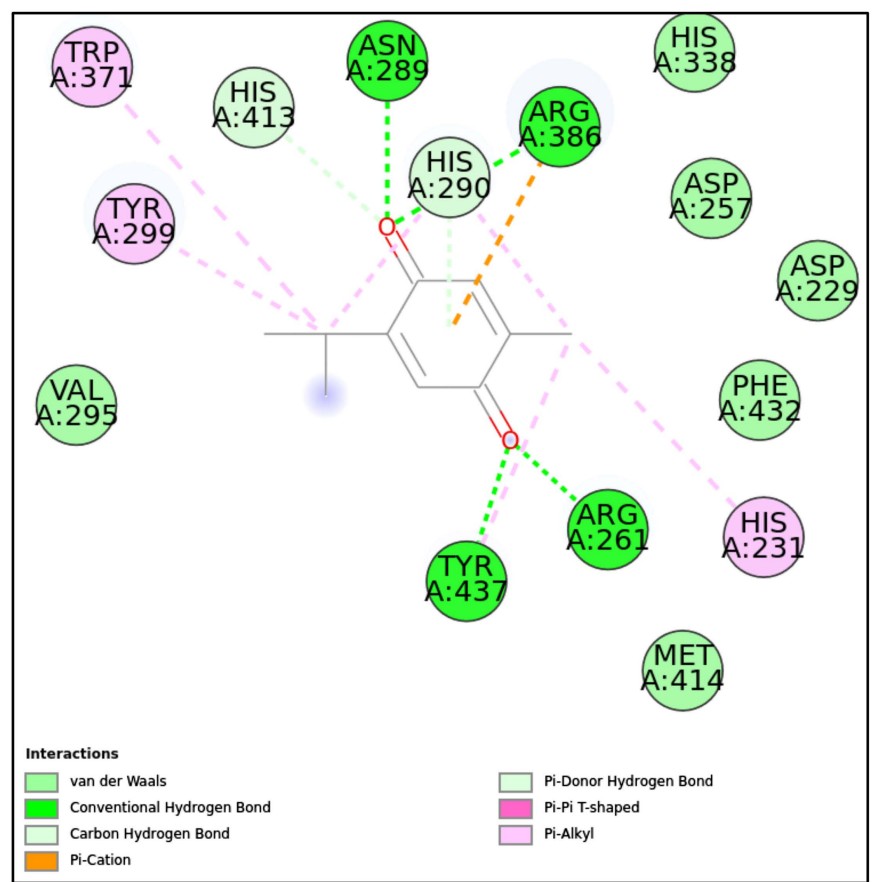

**Fig 12. 2D molecular interactions between TQ and 5JPF (PDB).** 2D molecular interactions between TQ and 5JPF, showing van der Waals forces, conventional and carbon-hydrogen bonds, π-cation, π-alkyl, and π-π T-shaped interactions.

with Asn154 suggests potential local steric hindrance, the comprehensive interaction profile underscores strong and stable ligand binding (Table 3 and S1 Fig.).

b) **TMP with 5UIV:** The interaction between TMP and the 5UIV protein demonstrates a highly stable and specific binding profile, supported by a combination of hydrogen bonds, covalent interactions, π-cation and π-alkyl interactions, salt bridges, and van der Waals forces. The key residue Arg39 forms a critical hydrogen bond, while Tyr100, Ser96, and Glu159 play crucial roles through covalent interactions. Arg92, Lys17, Lys35, Asp13, and Asp91 are involved in π-cation interactions and salt bridges, stabilizing the phosphate group of TMP. Furthermore, Phe67, Leu51, and Pro37 contribute through π-alkyl and alkyl interactions. Van der Waals interactions with Glu162, Tyr161, Gln167, Phe36, and Arg14 further stabilize the complex. These diverse interactions fine-tune TMP's position within the binding pocket, highlighting its potential as a molecular probe or a promising candidate for drug design targeting 5UIV (Table 3 and S2 Fig.).

c) **MLR with 5JPF**: The interaction between MLR and the 5JPF protein is characterized by a diverse network of interactions, including hydrogen bonds, alkyl, π-alkyl interactions, salt bridges, and van der Waals forces. Critical hydrogen bonds formed by Tyr299, Tyr437, and

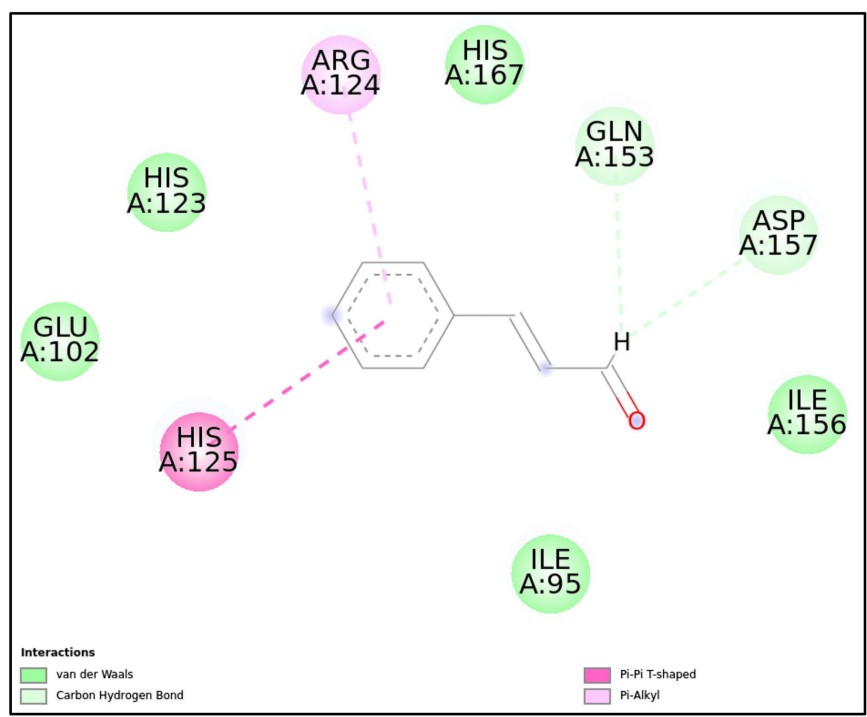

**Fig 13. 2D molecular interactions between CN and 1LMH (PDB).** 2D molecular interactions between CN and 1LMH, showing van der Waals forces, carbon-hydrogen bonds, π-alkyl, and π-π T-shaped.

Glu385 are pivotal in anchoring the ligand within the binding pocket. Attractive charge interactions and salt bridges involving His231, Arg261, and Arg386 further stabilize the ligand. Val388, Trp371, Cys438, and Val415 contribute to hydrophobic alkyl and π-alkyl interactions. Additional van der Waals interactions with Asn367, Val360, Pro361, Asp362, Asn294, Gly387, Val298, Val295, Asn289, His290, and Phe441 further enhance the stability and precise orientation of the ligand. This comprehensive interaction profile underscores the strong binding affinity between MLR and 5JPF, highlighting its potential as a candidate for structural and drug design studies (Table 3 and S3 Fig.).

d) **ACT with 1LMH:** The interaction between ACT and 1LMH is driven by a combination of hydrogen bonds, carbon-hydrogen bonds, π-alkyl interactions, and van der Waals forces. Arg124 forms a critical hydrogen bond, while His98 and Ile95 participate in carbon-hydrogen bonding. π-Alkyl interactions involve His125, contributing to hydrophobic stability. Additionally, Val122, Asp157, Gln153, Asn160, Ile156, Lys94, Val96, Ser97, Glu102, His123, and His167 are involved in van der Waals interactions, further stabilizing the ligand within the binding pocket. This diverse combination of interactions ensures a robust binding profile, making ACT a promising candidate for further drug design or structural studies targeting the 1LMH protein (Table 3 and S4 Fig).

e) **LIM with 1EQP:** The binding of LIM to 1EQP is characterized by a network of strong hydrogen bonds formed by residues Asn146, Asp145, Glu192, Arg312, Tyr153, Arg309, Asp151, Leu304, and Glu292, which stabilize the polar groups of the ligand. Carbon-hydrogen bonding involves Glu27 and Gly143, adding further stabilization. Van der Waals interactions with Tyr29, His135, Trp363, Tyr255, Trp373, Phe258, Asn305, Phe144, Arg150, Tyr317, and Phe229 contribute to the overall stability of the ligand in the active

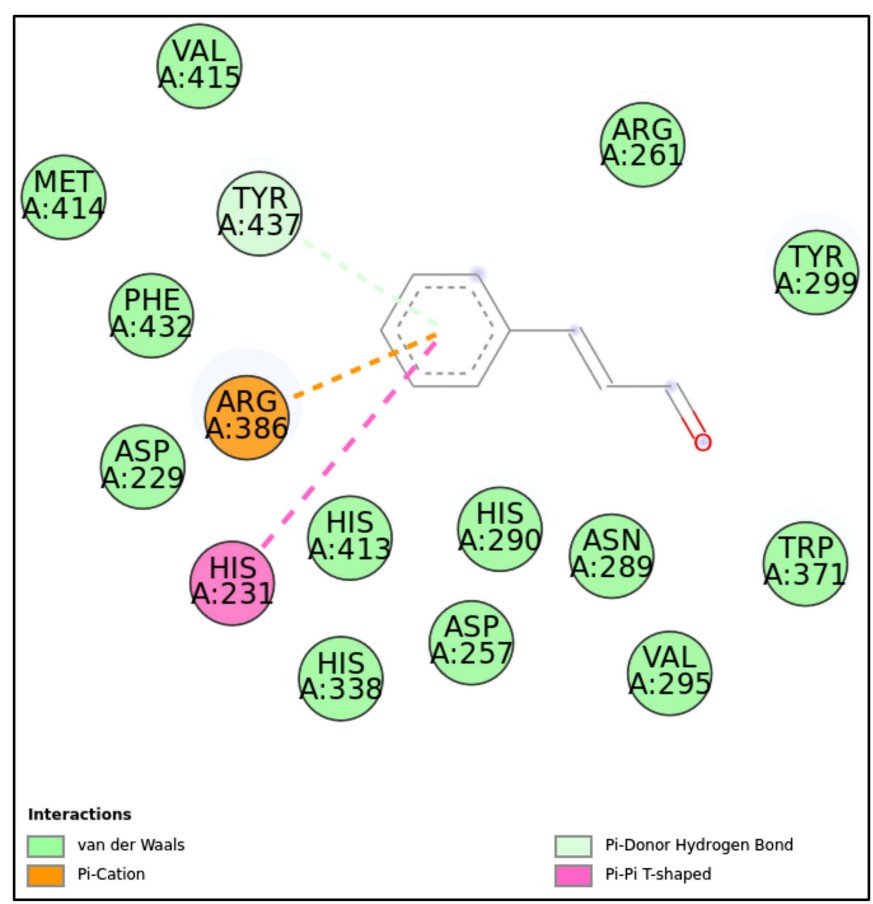

**Fig 14. 2D molecular interactions between CN and 1JPF (PDB).** 2D molecular interactions between CN and 1JPF, showing van der Waals forces, π-cation, π-π T-shaped interactions, and π-donor hydrogen bonds.

site. This comprehensive interaction profile highlights a stable binding configuration, positioning LIM as a strong candidate for further ligand-protein interaction studies targeting 1EQP (Table 3 and S5 Fig.).

## Discussion

### MDR pathogen

MDR pathogens have emerged as a significant global threat to public health, particularly in developing countries, owing to inadequate healthcare facilities and hygiene practices. These pathogens are characterized by their ability to resist the effects of several antibiotics, making them difficult to treat and increasing the risk of morbidity and mortality in infected individuals. The increase in MDR pathogens is attributed to the overuse and misuse of antibiotics, which leads to the selection of resistant strains. The horizontal transfer of resistance genes between different bacteria and the lack of new antibiotics exacerbate this problem. MDR pathogens can cause various infections, including bloodstream infections, pneumonia, urinary tract infections, and wound infections, and they are associated with increased healthcare costs and prolonged hospital stays [1,77]. Thus, there is an urgent need to identify new strategies for controlling the spread of MDR infections.

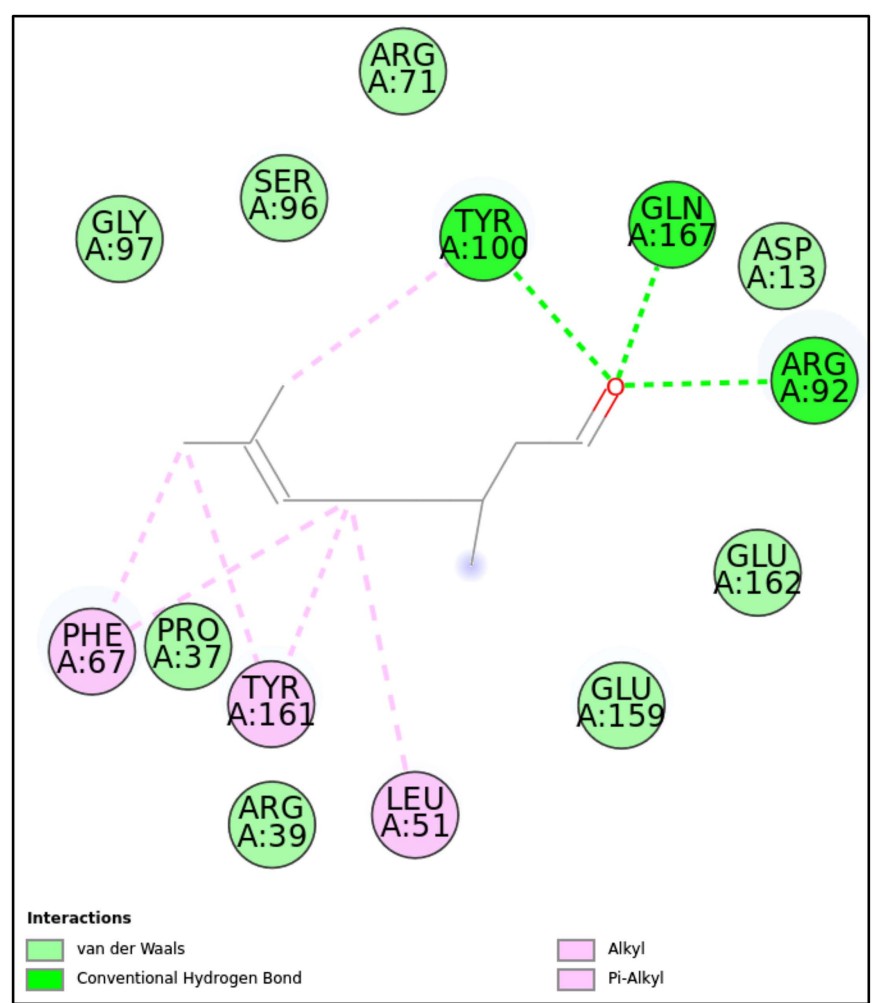

**Fig 15. 2D molecular interactions between CIT and 5UIV (PDB).** 2D molecular interactions between CIT and 5UIV, showing van der Waals forces, conventional hydrogen bonds, alkyl, and π-alkyl interactions.

## Importance of EOs and mechanisms of action

Many reports have suggested that plant-based products like EOs have substantial antimicrobial potential and could treat various human infections, including MDR [22,41,78]. EOs exert antibacterial and antifungal effects through various mechanisms that disrupt the normal functioning of microbial cells. These mechanisms include:

- **Cell Membrane Disruption**: EOs contain compounds such as terpenes and phenols that can interact with the lipid bilayer of microbial cell membranes, causing increased permeability. This leads to the leakage of essential cellular contents, ultimately resulting in cell death [79].

- **Interference with Enzymatic Activity**: Certain EO components can inhibit key enzymes involved in microbial metabolism. For instance, phenolic compounds such as EU (found in clove oil) can inhibit enzymes that are critical for cell wall synthesis or energy production, impeding the microorganism's ability to survive [79].

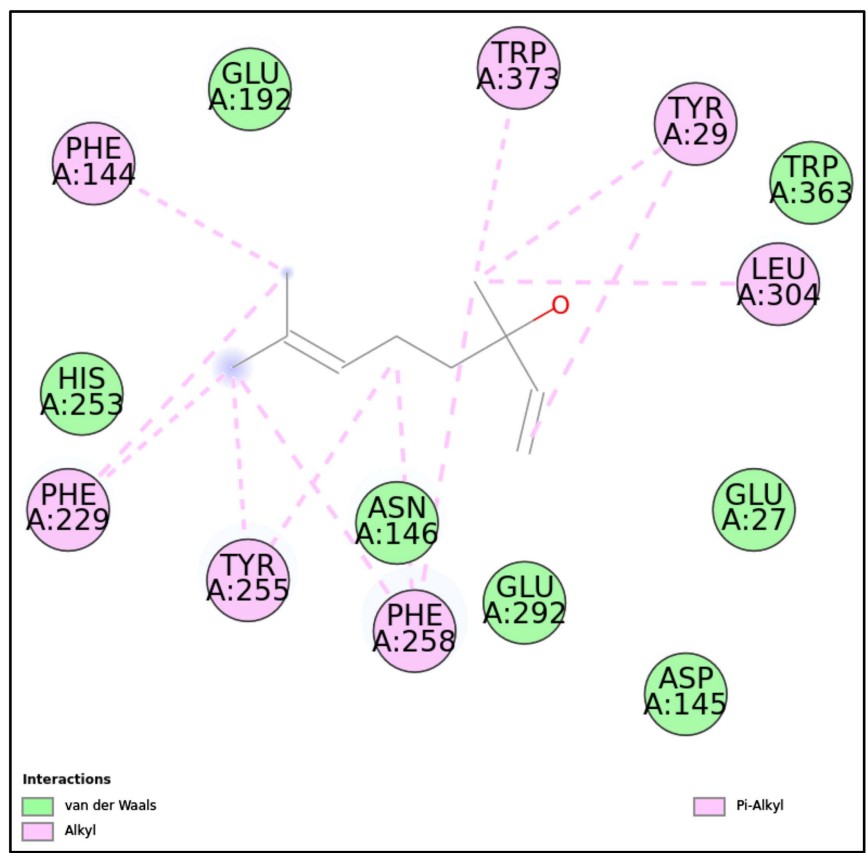

**Fig 16. 2D molecular interactions between LIN and 1EQP (PDB).** 2D molecular interactions between LIN and 1EQP, showing van der Waals forces, alkyl, and π-alkyl interactions.

- **Oxidative Stress Induction**: EOs can generate reactive oxygen species (ROS) within microbial cells, leading to oxidative stress. This damages proteins, lipids, and DNA, further contributing to the antimicrobial action [80].

- **Disruption of Quorum Sensing**: EOs may also interfere with quorum sensing, a process by which bacteria communicate and coordinate their activities, including biofilm formation. By disrupting this communication, EOs can prevent the establishment and maintenance of biofilms, which are protective layers that render bacteria more resistant to antibiotics [79].

These multifaceted mechanisms contribute to the broad-spectrum antimicrobial and anti-biofilm activities of EOs, making them promising agents against MDR pathogens.

## Purpose of the study

In this context, the current study investigated the *in vitro* antimicrobial and antibiofilm potential of four EOs—CBO, BSO, CNBO, and CTLO—against 19 selected pathogens, including MDR pathogens. Additionally, molecular docking simulations of five active ingredients (CN, EU, TQ, CIT, and LIN) of these EOs were performed against 23 microbial target proteins to validate their bioactivities. To validate the findings of this study, the antimicrobial and antibio-film activities of the tested EOs were compared with those reported in previous studies.

**Table 2. Ligand–protein interaction analyses for selected ligand-protein docked systems.**

| Target Protein with Selected Ligand | Residues Involved in Conventional H-Bonds | Residues Involved in Covalent Interactions (C-C, C-O, C-H, and C-N) | Residues Involved in Non-Covalent Interactions (Alkyl, Pi-Alkyl, Pi-Cation, Pi-Anion, Pi-Pi T Shaped, Pi-Sigma, Pi-Donor Hydrogen, and Pi-Sulphur) | Residues Involved in van der Waals Interactions |
|---|---|---|---|---|
| 5MM8-EU | Phe149 | Ser157 | Pro120, Val150, Val4, Tyr130, Ile116 | Glu131, Asn154, Val117, Gly118, Tyr119, Glu151, Glu1 |
| 5UIV-EU | Lys35 | Asp13 | Arg39, Glu159, Arg92, Tyr100 | Gly157, Gly155, Glu162, Tyr161, Ser96, Lys17, Pro37, Phe36, Phe67, Leu51 |
| 5MM8-TQ | Glu151, Asn154 | – | Pro120, Val4, Val150, Phe149 | Glu1, Ile116, Glu131, Ser132, Val117, Gly118, Tyr119, Ser157, Tyr130 |
| 5JPF-TQ | Asn289, Arg386, Arg261, Tyr437 | His290, His413 | Trp371, Tyr299, His231 | His338, Asp257, Asp229, Phe432, Met441 |
| 1LMH-CN | – | Gln153, Asp157 | Arg124, His125 | Glu102, His123, His167, Ile156, Ile95 |
| 5JPF-CN | – | Tyr437 | Arg386, His231 | Val415, Met414, Phe432, Asp229, His413, His290, Asn289, Trp371, His338, Asp257, Val295, Arg261, Tyr229 |
| 5UIV-CIT | Tyr100, Gln167, Arg92 | – | Phe67, Tyr161, Leu51 | Gly97, Ser96, Arg71, Asp13, Glu162, Glu159, Arg39, Pro37 |
| 1EQP-LIN | – | – | Phe144, Trp373, Tyr29, Leu304, Phe258, Tyr255, Phe229 | Glu192, Trp363, Glu27, Asp145, Glu292, Asn146, His253 |

**Table 3. Co-crystallized/standard ligand-protein interaction analyses for the selected proteins.**

| Target Protein with Co-Crystallized/Standard Ligand | Residues Involved in Conventional H-Bonds | Residues Involved in Covalent Interactions (C-C, C-O, C-H, and C-N) | Residues Involved in Non-Covalent Interactions (Alkyl, Pi-Alkyl, Pi-Cation, Pi-Anion, Pi-Pi T Shaped, Pi-Sigma, Pi-Donor Hydrogen, and Pi-Sulphur) | Residues Involved in van der Waals Interactions |
|---|---|---|---|---|
| 5MM8-BMD | Ser157, Tyr130 | – | Pro120, Val150, Val4 | Glu151, Ile116, Gly118, Val117, Tyr119, Glu131, Ser132, Lys5, Phe149 |
| 5UIV-TMP | Arg39 | Tyr100, Ser96, Glu159 | Pro37, Leu51, Phe67, Arg92, Lys35, Lys17, Asp13, Asp91 | Glu162, Tyr161, Gln167, Phe36, Arg14 |
| 5JPF- MLR | Tyr 299, Tyr437, Glu385 | – | Val388, Trp371, Cys438, Val415, His231, Arg386, Arg261, | Asn367, Val360, Pro361, Asp362, Asn294, Gly387, Val298, Val295, Asn289, His290, Phe441 |
| 1LMH-ACT | Arg124 | His98, Ile95 | His125 | His123, His167, Val122, Asp157, Gln153, Asn160, Ile156, Lys94, Val96, Ser97, Glu102 |
| 1EQP-LAM | Asn146, Asp145, Glu192, Arg312, Tyr 153, Arg309, Asp151, Glu292, Leu304 | Gly143, Glu27 | – | Tyr29, Phe229, Tyr317, Arg150, Trp373, Asn305, Phe144, Phe258, Try255, Trp363, His135 |

## Comparison to literature

Many studies have highlighted the substantial antimicrobial and antibiofilm potentials of CBO, BSO, CNBO, and CTLO against various human pathogens, including MDR strains [11,22,24,44,78,81–89]. For example, Gupta and Prakash reported that CBO exhibits notable antimicrobial activity against *Streptococcus mutans*, *Streptococcus salivarius*, *Lactobacillus* spp., *Bacillus* spp., *Micrococcus* spp., *S. aureus*, *Halobacterium* spp., *Veilonella* spp., *P. aeruginosa*, *Pseudomonas* spp., *A. niger*, *A. fumigatus*, *Aspergillus* sp., *Alternaria* sp., *Rhizomucor* sp., *Rhizopus* sp., and *Penicillium* sp., with mean ZIDs ranging from 9.0 to 19.0 mm for bacteria and 28.0 to 42.0 mm for fungi using the well-diffusion method with 50 μL oil/well [90].These findings align with our results, where CBO exhibited ZIDs ranging from 9.0 to 20.0 mm for

bacteria and 23.0 to 32.0 mm for fungi using the disc-diffusion method with 10 μL oil/disc. Similarly, Ahmad et al. demonstrated CBO's antifungal activity against *Candida albicans*, corroborating our findings [89]. For *E. coli*, Mejía-Argueta et al. reported a mean ZID of 15.59 mm using the disc-diffusion method with 10 μL oil/disc, comparable to our findings of a mean ZID of 14.0 ± 0.2 mm, along with MIC and MBC values of 3.125 μL/mL and 6.25 μL/mL, respectively [30].

Naveed et al. demonstrated that CBO and CNBO exhibit potent antimicrobial activity against *S. typhi*, *S. paratyphi*-A, *E. coli*, *S. aureus*, *Bacillus licheniformis*, and *Pseudomonas fluorescens* (*P. fluorescens*), with CNBO outperforming CBO [84]. These results corroborate our study, where CNBO was found to have superior antimicrobial activity against all tested pathogens, including *S. typhimurium* and *P. aeruginosa*. Similarly, Aumeeruddy-Elalfia et al. highlighted CNBO's efficacy against *Acinetobacter* sp., *K. pneumoniae*, *P. vulgaris*, *E. faecalis*, *S. aureus*, and *S. epidermidis*, with MIC values of 8.0 mg/mL, 2.0 mg/mL, 8.0 mg/mL, 4.0 mg/mL, 0.5 mg/mL, and 1.0 mg/mL, respectively, consistent with our findings [91]. Mohammed et al. reported that BSO has substantial activity against Gram-positive bacteria, including *S. aureus*, MRSA, *B. subtilis*, and *B. cereus*, with ZIDs ranging from 6.75 to 15.75 mm using 100 μL oil/well [31]. These findings align with our results, where BSO exhibited potent antimicrobial activity against Gram-positive bacteria, with ZIDs ranging from 17.0 to 46.0 mm using 10 μL oil/disc. Forouzanfar et al. reported that BSO has substantial antimicrobial activity against *S. aureus* (ATCC 29737), *E. coli* (ATCC 8739), and various species of plant fungi [44]. These results partially corroborated our results. [44]. Regarding CTLO, Burt reported its antimicrobial activity against *E. coli*, *S. typhimurium*, and *S. aureus*, with MIC values between 0.6 and 2.5 μL/mL [92]. In contrast, our study observed no antibacterial activity for CTLO but significant antifungal activity against *C. albicans* and *A. niger*, with MIC values of 0.78–1.56 μL/mL.

The *in vitro* MBIC and MBEC results demonstrated the strong antibiofilm potential of the tested EOs against all pathogens, highlighting their ability to prevent biofilm formation and eradicate established biofilms. This aligns with the molecular docking results, which revealed robust interactions between the bioactive compounds and biofilm-associated proteins. For example, EU's high binding affinity with 5MM8 (-10.48 kcal/mol) and 5UIV (-10.66 kcal/mol), driven by stabilizing hydrogen bonds and hydrophobic interactions, correlates with its superior *in vitro* antibiofilm activity. Similarly, CN exhibited strong docking interactions with 1LMH (-8.13 kcal/mol) and 5JPF (-7.11 kcal/mol), supporting its efficacy in biofilm disruption and eradication. TQ's robust binding with 5JPF (-9.09 kcal/mol) further validates its ability to target biofilm-forming bacteria effectively. In contrast, CIT and LIN demonstrated moderate docking affinities, relying on hydrophobic interactions, which align with their relatively lower antibiofilm activity observed *in vitro*. These findings provide a mechanistic explanation for the antibiofilm efficacy of these EOs, emphasizing the role of molecular interactions in their biofilm-targeting capabilities.

Regarding antibiofilm activity, Somrani et al. reported that CBO effectively inhibited biofilms of *Listeria monocytogenes (L. monocytogenes)* and *Salmonella enteritidis (S. enteritidis)* by up to 61.8% and 49.8%, respectively, at MIC concentrations (0.05 mg/mL for *L. monocytogenes* and 0.1 mg/mL for *S. enteritidis*) [93]. Similarly, our study demonstrated substantial antibiofilm activity for CBO against all tested pathogens, including Gram-positive and Gram-negative bacteria and fungi. For BSO, our study provides the first report of its antibiofilm activity against Gram-positive bacterial and fungal strains. Additionally, Jeong et al. reported CNBO's antibiofilm efficacy against multispecies oral biofilms at a 5% (v/v) concentration [94], corroborating our findings that CNBO exhibits robust antibiofilm activity across all tested organisms. In contrast, Guandalini et al. showed CTLO's antibiofilm activity against *S. aureus* and *C. albicans* [95], which partially aligns with our findings, as CTLO exhibited antibiofilm activity only against fungal strains in our study.

These results collectively highlight the broad-spectrum antimicrobial and antibiofilm potentials of CBO, BSO, and CNBO. However, CTLO's efficacy appears limited to fungal pathogens, suggesting its activity is more specific and dependent on its chemical composition. This specificity reinforces the importance of understanding the molecular mechanisms and chemical profiles of EOs to optimize their applications against various pathogens.

These observed activities can be further explained by the molecular docking results, which provide insights into the specific mechanisms of action of the active ingredients.

## Mechanistic insights and antifungal activity

The superior antimicrobial and antibiofilm activities of CNBO, CBO, and BSO can be attributed to their respective active ingredients, which showed strong molecular interactions with microbial targets.

- **CN** demonstrated high binding affinities with proteins like 1LMH (-8.13 kcal/mol) and 5JPF (-7.11 kcal/mol). These interactions were mediated by π-cation, π-π T-shaped, and van der Waals forces, which target microbial enzymatic activity and disrupt cell walls. CN's strong antibiofilm and antifungal activity are likely due to its ability to target fungal sterols like ergosterol and induce oxidative stress. This is supported by CNBO's potent activity, with MIC values as low as 0.195 μL/mL against fungal strains like *C. albicans* and *A. niger*.

- **EU** exhibited robust binding with 5MM8 (-10.48 kcal/mol) and 5UIV (-10.66 kcal/mol) through hydrogen bonds and hydrophobic interactions, correlating with its strong anti-biofilm activity. Its ability to disrupt quorum sensing and generate oxidative stress further supports its activity against biofilm-forming bacteria. CBO, which contains EU, showed substantial activity against Gram-positive and fungal strains, with MIC values ranging between 0.195–3.125 μL/mL.

- **TQ** showed strong binding to 5JPF (-9.09 kcal/mol) and 5MM8 (-9.32 kcal/mol), mediated by hydrogen bonds and van der Waals forces. Its ability to interact with biofilm-associated proteins and disrupt microbial metabolic pathways explains BSO's efficacy, particularly its potent antibiofilm activity reported for the first time in this study.

- **CIT** demonstrated moderate binding affinities with 5UIV (-9.43 kcal/mol) through hydrogen bonds with key residues like Tyr100 and Arg92. Its relatively weaker activity against Gram-negative bacteria and biofilms aligns with its dependence on hydrophobic interactions, which are less effective against bacteria with protective outer membranes.

- **LIN** showed weaker binding with proteins like 1EQP (-8.89 kcal/mol), relying on hydrophobic and van der Waals interactions. Its moderate antibiofilm activity corresponds to its inability to form strong hydrogen bonds or covalent interactions, limiting its efficacy compared to CN and EU.

These mechanistic insights connect the observed bioactivities of the EOs to the molecular interactions of their active ingredients, highlighting their potential for combating MDR pathogens and biofilm-associated infections.

## Gram-negative bacteria susceptibility

Gram-negative bacteria were generally less susceptible to the tested EOs than Gram-positive bacteria. This reduced susceptibility can be attributed to the protective outer membrane and lipopolysaccharide (LPS) layer in Gram-negative bacteria, which act as barriers to the penetration of hydrophobic EO components. For instance, while BSO exhibited potent activity against Gram-positive bacteria, its efficacy against Gram-negative pathogens was lower.

Similarly, CTLO lacked antibacterial activity entirely but demonstrated significant antifungal activity, further emphasizing the role of membrane composition in determining susceptibility.

### Advancing knowledge

The study effectively integrates *in vitro* antimicrobial and antibiofilm activity data with molecular docking insights, providing mechanistic explanations for the observed efficacy of EOs. The notable antimicrobial and antibiofilm activities of CNBO, BSO, and CBO highlight their strong potential as alternative therapeutic agents for combating MDR pathogens and biofilm-associated infections, with CTLO showing promise against fungal pathogens.

### Practical applications

The findings suggest several practical applications for these EOs. Their broad-spectrum antimicrobial and antibiofilm activities make them suitable for use in:

- **Healthcare Settings:** As antimicrobial coatings for medical devices, wound care formulations, or disinfectant solutions to combat MDR pathogens and biofilm-associated infections.
- **Pharmaceutical Industry:** Development of antifungal therapies or natural preservatives for pharmaceutical products.
- **Food and Agriculture:** Preserving food products and treating fungal plant infections.

These findings demonstrate the efficacy of EOs as potential alternatives to synthetic antimicrobials and highlight their versatility in addressing global challenges posed by MDR infections in healthcare and agriculture.

## Conclusions

In conclusion, the emergence of MDR pathogens has become a global public health threat, necessitating the search for novel and effective antimicrobial agents. This study evaluated the *in vitro* and *in silico* antimicrobial and antibiofilm potentials of EOs derived from *S. aromaticum L.*, *N. sativa L.*, *C. zeylanicum*, and *P. citrosum* against selected human pathogens, including MDR pathogens. These findings suggest that these EOs have promising antimicrobial and antibiofilm activities, with the highest activity observed for CBO, BSO, and CNBO against *S. epidermidis*. In contrast, CNBO exhibited the highest antifungal activity against the tested fungi. Molecular docking results further validated the antimicrobial potential of these EOs, highlighting their ability to interact with the target proteins of several pathogens.

The translational potential of these EOs lies in their utility as alternative or complementary therapies for tackling MDR infections, particularly MDR biofilms, which are difficult to treat with conventional antibiotics. Their broad-spectrum activity underscores their applicability in healthcare, pharmaceutical, and agricultural settings.

### Limitations

While these findings are promising, this study has some limitations. Specifically, we could not test the synergistic effects between EOs, which could provide additional insights into their combined efficacy. Furthermore, the lack of *in vivo* validation limits the direct clinical applicability of these results. Future research should address these limitations by exploring EO synergism and conducting *in vivo* studies to confirm safety and efficacy.

Overall, the findings of this study offer valuable insights into the potential use of EOs as alternative or complementary therapies for MDR infections, highlighting the need for further research to explore their translational potential fully.

## Supporting information

**S1 Fig. Interactions between BMD (ligand) and 5MM8 (protein).**
(TIF)

**S2 Fig. Interactions between TMP (ligand) and 5UIV (protein).**
(TIF)

**S3 Fig. Interactions between MLR (ligand) and 5JPF (protein).**
(TIF)

**S4 Fig. Interactions between ACT (ligand) and 1LMH (protein).**
(TIF)

**S5 Fig. Interactions between LIM (ligand) and 1EQP (protein).**
(TIF)

**S1 Table. Proteins (enzymes) selected for *in silico* molecular docking studies of active ingredients from CBO, BSO, CNBO, and CTLO targeting bacterial and fungal pathways.**
(PDF)

**S2 Table. One-way ANOVA of antimicrobial activity of essential oils against tested microorganisms.**
(PDF)

**S3 Table. Post Hoc analysis of antimicrobial activity of essential oils against tested microorganisms.**
(PDF)

**S4 Table. The selected target proteins and their standard/co-crystallized ligands.**
(PDF)

## Author contributions

**Conceptualization:** Kamal Ahmad Qureshi.

**Data curation:** Kamal Ahmad Qureshi, Humaira Ismatullah.

**Formal analysis:** Kamal Ahmad Qureshi, Adil Parvez, Humaira Ismatullah, Hanan Almahasheer.

**Funding acquisition:** Osamah Al Rugaie.

**Investigation:** Kamal Ahmad Qureshi, Adil Parvez, Humaira Ismatullah, Hanan Almahasheer.

**Methodology:** Kamal Ahmad Qureshi, Humaira Ismatullah.

**Project administration:** Kamal Ahmad Qureshi.

**Software:** Kamal Ahmad Qureshi, Adil Parvez, Humaira Ismatullah.

**Supervision:** Kamal Ahmad Qureshi.

**Validation:** Kamal Ahmad Qureshi, Adil Parvez, Humaira Ismatullah, Osamah Al Rugaie.

**Visualization:** Kamal Ahmad Qureshi, Adil Parvez, Humaira Ismatullah.

**Writing – original draft:** Kamal Ahmad Qureshi, Humaira Ismatullah.

**Writing – review & editing:** Kamal Ahmad Qureshi, Adil Parvez, Humaira Ismatullah, Hanan Almahasheer, Osamah Al Rugaie.

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
