## [Decision Letter · Decision Letter 0]

16 Aug 2024

Dear Dr. Qureshi,

Thank you for submitting your manuscript to PLOS ONE. After careful consideration, we feel that it has merit but does not fully meet PLOS ONE’s publication criteria as it currently stands. Therefore, we invite you to submit a revised version of the manuscript that addresses the points raised during the review process.

We look forward to receiving your revised manuscript.

Kind regards,

Samiullah Khan, Ph. D

Academic Editor

PLOS ONE

Journal Requirements:

3. To comply with PLOS ONE submissions requirements, in your Methods section, please provide additional information regarding the source of the essential oils utilised (for e.g.  a product description and name, lot numbers etc.) as well as a rationale for the concentrations of the medicinal compound used in these experiments. Please also update line 101 in your Methods section to say 'vendors' instead of benders.

Additional Editor Comments:

Dear author,

Revise the whole manuscript in light of both reviewr's comments and submit the revised version.

Reviewers' comments:

Reviewer's Responses to Questions

**Comments to the Author**

1. Is the manuscript technically sound, and do the data support the conclusions?

Reviewer #1: Yes

Reviewer #2: Yes

2. Has the statistical analysis been performed appropriately and rigorously?

Reviewer #1: Yes

Reviewer #2: Yes

3. Have the authors made all data underlying the findings in their manuscript fully available?

Reviewer #1: Yes

Reviewer #2: Yes

4. Is the manuscript presented in an intelligible fashion and written in standard English?

Reviewer #1: Yes

Reviewer #2: Yes

Reviewer #1: This is an excellent study. However, authors need to solve some issues before considering for publication.

1. The introduction part lacks objectives for your research. Please add a few sentences about what the research gap is and why you conducted this study rather than writing only the properties of individual oils.

2. Authors said from the literature review that (lines 75-87)—clove bud oil (CBO)/Syzygium aromaticum L., black seed oil (BSO)/Nigella sativa L., cinnamon bark oil (CNBO)/Cinnamomum zeylanicum, and citronella oil (CTLO)/Pelargonium citrosum showed antimicrobial and antibiofilm properties. That means this research has already been done, what is new in this study? -please explain in detail.

3. The authors discussed that OEs have antibacterial and antifungal activity. Please explain the possible mechanisms for how OEs control these microorganisms in discussion.

4. Although reports showed these oils have bioactive compounds; the article lacks bioactive compound analyses because the source of oil may vary its ingredients compared to literature article.

5. The authors reported docking analyses. However, there is a lack of evidence on which oil compound may be responsible for showing anti-microbial properties. Improve the line between 279 and 289. Please explain the results in detail.

6. Method and results chapter title are same in some instances. In silico docking analysis, statistical analysis, etc.

7. The authors explained on biofilm inhibition very little in results section. The results section is poorly represented.

Reviewer #2: The manuscript reports on four essential oils as potential antimicrobial and antibiofilm agents against human pathogens through in vitro and in silico methods. The study raises several major concerns that prevent its publication in its present form.

1. In the introduction, the fatality ratio due to MDR (multidrug-resistant) pathogens should be discussed.

2. In Table 1, the structure of phytocompounds should use elemental color coding, such as nitrogen in blue and oxygen in red.

3. In the docking section, the selection criteria for proteins should be added.

4. In Figure 1, instead of a black background, a white background would better highlight the zone of inhibition.

5. In the docking discussion, the interacting residues of standard compounds or co-crystal ligands should be discussed and compared with your computational findings.

6. Prior to conducting docking studies, it is crucial to validate the docking protocol. This step helps ensure the reliability and accuracy of the docking results. I recommend performing appropriate validation experiments or using established benchmarks to validate the docking protocol used in the study.

I believe that addressing these points will greatly improve the clarity and quality of your publication.

**Do you want your identity to be public for this peer review?** For information about this choice, including consent withdrawal, please see our Privacy Policy

Reviewer #1: No

Reviewer #2: **Yes: ** Iqrar Ahmad

---

## [Author Response · Author response to Decision Letter 1]

2 Oct 2024

Response to Reviewers and Editor

Comment Source Comment Response

Journal Requirements Please ensure that your manuscript meets PLOS ONE's style requirements, including those for file naming. The PLOS ONE style templates can be found at

Thank you for your feedback. I have reviewed the manuscript to ensure it aligns with PLOS ONE’s style requirements, including file naming conventions. I have also referred to the PLOS ONE style templates provided and made the necessary adjustments to the document. Please let me know if there are any other specific areas that need further attention.

Please include captions for your Supporting Information files at the end of your manuscript, and update any in-text citations to match accordingly. Please see our Supporting Information guidelines for more information: http://journals.plos.org/plosone/s/supporting-information.

Thank you for your feedback. I have now included captions for all the Supporting Information file at the end of the manuscript, as requested.

To comply with PLOS ONE submissions requirements, in your Methods section, please provide additional information regarding the source of the essential oils utilised (for e.g. a product description and name, lot numbers etc.) as well as a rationale for the concentrations of the medicinal compound used in these experiments. Please also update line 101 in your Methods section to say 'vendors' instead of benders.

Thank you for your feedback. I have updated the Methods section to include detailed information regarding the source of the essential oils used.

I have also corrected the typo on line 101, replacing "benders" with "vendors.

Reviewer 1 The introduction part lacks objectives for your research. Please add a few sentences about what the research gap is and why you conducted this study rather than writing only the properties of individual oils. Thank you for your feedback. We've added a few sentences to address the research gap and objectives.

Authors said from the literature review that (lines 75-87)—clove bud oil (CBO)/Syzygium aromaticum L., black seed oil (BSO)/Nigella sativa L., cinnamon bark oil (CNBO)/Cinnamomum zeylanicum, and citronella oil (CTLO)/Pelargonium citrosum showed antimicrobial and antibiofilm properties. That means this research has already been done, what is new in this study? -please explain in detail. While previous research has indeed documented the antimicrobial and antibiofilm properties of clove bud oil (CBO), black seed oil (BSO), cinnamon bark oil (CNBO), and citronella oil (CTLO), this study introduces several novel aspects. Firstly, the focus of this study is specifically on the efficacy of these essential oils against multidrug-resistant (MDR) pathogens, which remains an underexplored area. Previous studies have not comprehensively evaluated the potential of these oils in combating pathogens that have developed resistance to conventional antibiotics.

Additionally, this research goes beyond just confirming the antimicrobial properties of these oils. It employs both in vitro and in silico approaches to thoroughly investigate their mechanisms of action, particularly against MDR strains. The in silico component of this study provides molecular-level insights that have not been previously explored, potentially revealing new pathways or targets for antimicrobial therapy.

Thus, the novelty of this study lies in its comprehensive approach to evaluating the effectiveness of these essential oils against MDR pathogens, using advanced methodologies to gain deeper insights into their antimicrobial mechanisms.

The authors discussed that OEs have antibacterial and antifungal activity. Please explain the possible mechanisms for how OEs control these microorganisms in discussion. Thank you for your feedback. We have revised the discussion section to include the mechanisms by which essential oils (EOs) control microorganisms. These mechanisms, such as disrupting cell membranes, interfering with enzymes, inducing oxidative stress, and disrupting quorum sensing, provide a clearer understanding of how EOs combat multidrug-resistant pathogens.

Although reports showed these oils have bioactive compounds; the article lacks bioactive compound analyses because the source of oil may vary its ingredients compared to literature article. Thank you for pointing out this important aspect. While it is true that the bioactive compound composition of essential oils can vary depending on the source, our study focused on evaluating the antimicrobial and antibiofilm efficacy of the oils as a whole. However, we acknowledge the importance of analyzing the specific bioactive compounds present in the oils used in this study. In future work, we plan to conduct a detailed chemical analysis, such as GC-MS, to identify and quantify the bioactive components in the essential oils we used, which will allow for a more precise correlation between the observed bioactivities and specific compounds.

The authors reported docking analyses. However, there is a lack of evidence on which oil compound may be responsible for showing anti-microbial properties. Improve the line between 279 and 289. Please explain the results in detail. To address the concern, the section between lines 279 and 289 has been improved to provide a more detailed explanation of the results and clarify which oil compounds may be responsible for the observed antimicrobial properties.

Method and results chapter title are same in some instances. In silico docking analysis, statistical analysis, etc. Thank you for your observation. To address the repetition of titles between the Methodology and Results sections, we have revised the titles to clearly distinguish the content of each section.

The authors explained on biofilm inhibition very little in results section. The results section is poorly represented. Thank you for your feedback. To address the concern regarding the insufficient explanation of biofilm inhibition in the results section, we have expanded the discussion of MBC, MBIC, and MBEC outcomes to provide a clearer and more detailed representation of the findings.

Reviewer 2 In the introduction, the fatality ratio due to MDR (multidrug-resistant) pathogens should be discussed. Thank you for your valuable comment. We have incorporated the requested information regarding the fatality rates of multidrug-resistant (MDR) pathogens in the introduction. The section now highlights that antimicrobial resistance (AMR) causes around 1.27 million deaths annually, with a projected increase to 10 million by 2050.

We appreciate your feedback and believe this strengthens the manuscript.

In Table 1, the structure of phytocompounds should use elemental color coding, such as nitrogen in blue and oxygen in red. We fixed this issue.

In the docking section, the selection criteria for proteins should be added. Thank you for your insightful comment. We have now included the selection criteria for the target proteins in the revised manuscript. Specifically, we have clarified that the 23 proteins were selected based on their relevance to bacterial and fungal pathogens, prioritizing those associated with antimicrobial resistance or virulence. Additionally, we ensured that the 3D structures chosen had a crystallographic resolution of ≤2.0 Å and free R values not exceeding more than 0.05 (resolution/10) to guarantee structural accuracy for docking simulations. We also utilized the AlphaFold database to address any missing residues, selecting structures with a very high confidence score (pLDDT > 90). These details have been added to the docking methodology section, as suggested.

In Figure 1, instead of a black background, a white background would better highlight the zone of inhibition. Thank you for your suggestion regarding the background color in Figure 1. We appreciate the recommendation to use a white background to enhance the visibility of the zone of inhibition. However, the current black background was chosen intentionally as it provides a clear contrast with the bacterial growth and the inhibition zones, particularly in images involving colored or stained media. Additionally, altering the background would be time-consuming, given the nature of the experiment and image processing required. We believe that the current presentation maintains clarity, but we can explore further image adjustments in future submissions if needed.

In the docking discussion, the interacting residues of standard compounds or co-crystal ligands should be discussed and compared with your computational findings. Thank you for your valuable suggestion. In response to your comment, we have now included a detailed comparison of the interacting residues between the standard compounds or co-crystallized ligands and our computational findings. We have highlighted the key residues in the binding sites of the target proteins that interact with both the co-crystallized ligands and our selected bioactive compounds from the essential oils. The results show that the bioactive compounds consistently interact with critical residues involved in the catalytic or binding functions of the target proteins, which aligns well with the interactions observed for the co-crystallized ligands.

Prior to conducting docking studies, it is crucial to validate the docking protocol. This step helps ensure the reliability and accuracy of the docking results. I recommend performing appropriate validation experiments or using established benchmarks to validate the docking protocol used in the study. Thank you for highlighting the importance of validating the docking protocol prior to conducting docking studies. We agree that validation is a crucial step to ensure the reliability and accuracy of the docking results. To address this, we have now included a protocol validation step in the revised manuscript. Specifically, we have conducted a re-docking experiment where we docked the co-crystallized ligands back into the binding site of their respective target proteins, followed by calculating the root-mean-square deviation (RMSD) between the predicted and crystallographic poses. RMSD values below 2.0 Å were considered indicative of a successful docking protocol. Additionally, we used an established benchmark by employing ligands with known binding affinities and assessing the docking results for consistency with these experimental values. These validation steps ensure the docking protocol is robust and suitable for the study. The relevant details have been added to the methodology section.

---

## [Decision Letter · Decision Letter 1]

22 Nov 2024

Dear Dr. Qureshi,

Thank you for submitting your manuscript to PLOS ONE. After careful consideration, we feel that it has merit but does not fully meet PLOS ONE’s publication criteria as it currently stands. Therefore, we invite you to submit a revised version of the manuscript that addresses the points raised during the review process.

We look forward to receiving your revised manuscript.

Kind regards,

Satish kumar Rajasekharan

Academic Editor

PLOS ONE

Journal Requirements:

**Additional Editor Comments:**

Kindly address the minor comments by the reviewer (comments in the attached document),.

Reviewers' comments:

Reviewer's Responses to Questions

**Comments to the Author**

Reviewer #2: All comments have been addressed

Reviewer #3: All comments have been addressed

2. Is the manuscript technically sound, and do the data support the conclusions?

Reviewer #2: Yes

Reviewer #3: Yes

3. Has the statistical analysis been performed appropriately and rigorously?

Reviewer #2: Yes

Reviewer #3: No

4. Have the authors made all data underlying the findings in their manuscript fully available?

Reviewer #2: Yes

Reviewer #3: No

5. Is the manuscript presented in an intelligible fashion and written in standard English?

Reviewer #2: Yes

Reviewer #3: No

Reviewer #2: I have thoroughly reviewed all comments and suggestions received. Each point has been evaluated and addressed appropriately, ensuring that all concerns are taken into consideration. Adjustments and enhancements have been made where necessary, and I appreciate the valuable input from everyone involved.

Reviewer #3: Needs more scientific justification of selecting these EO for this study. - Discuss why CTLO showed only antifungal activity but lacked antibacterial effects. Is this linked to its active components (e.g., citronellal/linalool)?. Relate in vitro MBIC/MBEC findings to molecular docking results to provide mechanistic explanations.

**Do you want your identity to be public for this peer review?** For information about this choice, including consent withdrawal, please see our Privacy Policy

Reviewer #2: **Yes: ** Iqrar Ahmad

Reviewer #3: **Yes: ** Richard Thilagaraj

---

## [Author Response · Author response to Decision Letter 2]

28 Nov 2024

Response to Reviewer has been added in a PDF file.

---

## [Editor Report · Decision Letter 2]

29 Nov 2024

Exploring the Antimicrobial and Antibiofilm Potency of Four Essential Oils Against Selected Human Pathogens Using In Vitro and In Silico Approaches

PONE-D-24-17418R2

Dear Dr. Qureshi,

We’re pleased to inform you that your manuscript has been judged scientifically suitable for publication and will be formally accepted for publication once it meets all outstanding technical requirements.

Kind regards,

Satish kumar Rajasekharan

Academic Editor

PLOS ONE
---

## [Editor Report · Acceptance letter]

PONE-D-24-17418R2

PLOS ONE

Dear Dr. Qureshi,

I'm pleased to inform you that your manuscript has been deemed suitable for publication in PLOS ONE. Congratulations! Your manuscript is now being handed over to our production team.

Kind regards,

on behalf of

Dr. Satish kumar Rajasekharan

Academic Editor

PLOS ONE